# An Improved Model of Shade-affected Stream Temperature in Soil & Water Assessment Tool

Efrain Noa-Yarasca[1], Meghna Babbar-Sebens[1], Chris Jordan[2]

[1]School of Civil & Construction Engineering, Oregon State University, Corvallis, OR 97331.

[2]Conservation Biology Division, NWFSC, National Marine Fisheries Service, National Oceanic and Atmospheric Administration, 2032 SE OSU Dr., Newport, OR, 97365 USA.

*Correspondence to*: Efrain Noa-Yarasca (enoay7@yahoo.com)

**Abstract.** Stream temperatures have been increasing worldwide, in some cases, reaching unsustainable levels for aquatic life. Riparian re-vegetation has been identified as a strategy for managing stream temperatures by blocking direct solar radiation. In this study, the effects of riparian vegetation on stream temperatures were included within the Soil Water Assessment Tool (SWAT) model through a shade factor parameter. An equilibrium temperature approach was used to integrate the shade factor in an energy balance context. The stream temperature sub-model was improved using the new energy balance equation and integrated into SWAT. Unlike existing models, the modified SWAT model developed enables improved representation of two processes - mass and heat transfer - that influence stream temperature change and enables simulation of shading and its effects on stream temperatures at sub-basin scales. The updated SWAT model was tested in Dairy McKay Watershed, OR, USA, for four scenarios: current conditions of riparian vegetation, full restoration, efficient restoration, and no vegetation. The model calibration under current riparian vegetation showed good performance (NSE>0.74). Stream temperature reduction and number of days with stream temperatures above survival limits (NDSTASL) for aquatic species were also evaluated as measures of riparian shade performance. Findings showed average temperature reductions of 0.91 °C (SD = 0.69 °C) and reductions in NDSTASL of 17.1 days over a year for full riparian restoration, and average reductions of 0.86 °C (SD = 0.67 °C) and 16.2 days for efficient restoration. Notwithstanding the similar benefits, efficient restoration was 14.4% cheaper than full riparian vegetation restoration.

## 1 Introduction

Stream temperature is an important parameter in water quality not only because it is one of the main indicators of biodiversity and sustainable aquatic ecosystems in rivers, but also because it is directly linked to other water quality parameters such as dissolved oxygen, salinity, and pH (Brown, 1972; Poole & Berman, 2001; Risley et al., 2003; Winfree et al., 2018). Ranges in stream temperature determine the habitat suitability for aquatic species. Significant changes outside the natural ranges in stream temperature can cause the death or migration of endemic species and the potential entry of non-native species, leading to an ecological imbalance(Albertson et al., 2018; Isaak et al., 2012; Nelitz et al., 2007). For example, elevated stream temperatures can increase the solubility of certain heavy metals such as cadmium, zinc, and ammonia which are toxic for aquatic life(White

et al., 2017). High stream temperatures are also linked to low levels of dissolved oxygen, increases in conductivity, low levels of oxidation-reduction potential, decreases in pH, all of which can alter aquatic life and its viability (Fondriest Environmental Inc., 2014). Changes in water temperature also influence hydrological parameters such as evaporation through altering the heat flux at the air-water interface, as well as other parameters indirectly, because all processes in the water cycle are linked (Edinger

et al., 1974). Historical records from the past 30 to 100 years show that stream temperatures throughout the United States have significantly increased at rates of 0.009 to 0.077 °C/year (Kaushal et al., 2010). Unusual increases in water temperatures observed in the western US have exceeded limits for survival of certain aquatic species (Sherwood, 2015). For example, in the summer of 2015, the Oregon Department of Fish and Wildlife estimated an approximately 55% reduction in the sockeye salmon population along the lower Columbia River stretch due to stream temperature rising to 24.5 °C (Nguyen, 2021;

Sherwood, 2015). Over the past 70 years, the abundance of species such as Coho salmon has shown a drastic decline in California, with similar but less drastic trends in Oregon, due to various factors including elevated stream temperatures (NMFS, 2012, 2014). In the local area, Winter Steelhead, Coho Salmon, and resident Cutthroat Trout are among the primary inhabitants of the Dairy McKay watershed streams, whose population is declining due to a variety of water quality factors, including water temperature (CWL, 2019; Hennings, 2014; ODA, 2018). In this regard, the Oregon Plan identified salmon health as a crucial

indicator of the ecosystem (Hawksworth, 1999; ODEQ, 2001, 2008, 2010). Additionally, in this area, declines in ecosystem structure and function have also been linked to declines in salmon numbers(Hennings, 2014; ODA, 2018).

Changes in stream temperature are driven (i) by heat transfer processes that involve the gain/loss of heat in the water body by several thermodynamic pathways, and (ii) by mass transfer processes that involve the gain/loss of heat from hydrologic flows that interact and mix with the target stream (Boyd & Kasper, 2003; Chen, Carsel, et al., 1998; Chen, McCutcheon, et al., 1998).

Within these two types of processes, many factors corresponding to the channel morphology, hydrology, and vegetation surrounding the river affect the surface water temperature (Boyd, 1996; Risley et al., 2003; Winfree et al., 2018). These processes can also be influenced by human activities such as the discharge of industrial effluents with high temperatures, riverbed modifications, and alteration of the riverside vegetation favouring a greater solar exposure of the water body (Hester & Doyle, 2011; Poole & Berman, 2001). While warm flow discharges from industrial effluents are the main point source of

heat, short wave radiation is the main diffuse source of heat that alters stream temperatures (Boyd & Kasper, 2003; Poole & Berman, 2000). A reduction of riparian vegetation cover can increase loading of direct solar radiation on the body of water. On the other hand, reforestation of riparian vegetation can block much of this energy before reaching the surface of the stream, thereby, helping to maintain a relatively cool stream temperature (Abbott G., 2002; Fuller et al., 2022; LeBlanc & Brown, 2000). To illustrate, studies conducted on the Salmon River in northern California by Bond et al. showed that simulations of

partial riparian reforestation would reduce stream temperatures by 0.11 to 0.12 °C/km and full reforestation by 0.26 to 0.27 °C/km (Bond et al., 2015).

The increase in the temperature of streams in recent decades has stimulated the interest of researchers to study and establish predictive models. These models mainly classified as mechanistic or statistical, vary from simple to complex, involving few

to numerous parameters, with time scales ranging from minutes to months, and spatial scales ranging from local to global.

Mechanistic models are physics-based numerical models involving concepts of hydrological and energy balance processes in their equations, while statistical models are models that employ data-driven techniques to establish functional relationships between stream temperature and meteorological or physical parameters of the basin (Sohrabi et al., 2017; Stefan & Preud'homme, 1993). Although statistical models may yield reliable outcomes with few parameters and simple equations (Benyahya et al., 2007; Mohseni & Stefan, 1999), they often do not consider the right physical structures that characterize the

hydrological process and do not take into account the proper interaction of the hydrological process variables (Boyd & Kasper, 2003; Kim & Chapra, 1997).

Mechanistic models involve heat and mass transfer processes in their structure. Full heat transfer processes involve fluxes through the air-water interface, the water-sediment interface along the riverbed, and chemical reactions in the aquatic environment (Boyd & Kasper, 2003). However, few models have the capability to include a complete balance of heat input

and output in the stream temperature simulation. Rates of gain/lost heat from aquatic chemical reactions and through the water-sediment interface are often very small compared to heat fluxes through the air-water interface (Hebert et al., 2011). The mass transfer process requires establishing the inlet and outlet discharge flows through the water body boundaries and their corresponding temperatures. This involves knowing components from a hydrological model such as the stream tributary flows, the lateral flow, the outgoing or incoming flow rate of the groundwater, the precipitation that falls directly on the stream, and

the hyporheic exchange flow (Ficklin et al., 2012; Rothwell, 2005). For example, the *Heat Source* model integrates these heat and mass transfer processes into a river-scale analytical model (Boyd, 1996; Boyd & Kasper, 2003). The *i-Tree Cool River Model* is a 1D model that simulates the stream temperature including the advection, dispersion, energy flux and mixing processes on a river scale (Abdi et al., 2020; Abdi & Endreny, 2019). Previous works to integrate the heat transfer process into sub-basin-scale hydrologic models have resulted in models limited to certain regions and parameters, such as the Hydrologic

Simulation Program-FORTRAN (Chen, Carsel, et al., 1998), the Stream Network Temperature (SNTEMP) energy-balance-based model (Krause et al., 2005), the Distributed Hydrology Soil Vegetation Model (DHSVM) (Battin et al., 2007; Wigmosta , M.S. et al., 1994; Yearsley, 2009), and the Hydrodynamic and Water Quality model (CE-QUAL-W2) (Zhu et al., 2019).

In the same vein, Ficklin et al. (2012) developed a hydroclimatological stream temperature model (called "Ficklin model", here on), within the integrated watershed model, Soil and Water Assessment Tool (SWAT; (Arnold et al., 1998; Neitsch et al.,

2009), which involves simplified terms representing the mass transfer process and a surrogate term representing the heat transfer process. In the mass transfer process, the model follows a mixing approach of inflows and outflows into and out of rivers (snowmelt flow, surface runoff, lateral flow, groundwater flow - these are all SWAT model outcomes) associated with their corresponding temperatures, while in the heat transfer process it is represented by the difference between the air and water temperatures at the air-water interface multiplied by a calibrated coefficient.

Stream temperature simulations, conducted using the Ficklin et al. model in several watersheds in the Columbia River basin in Northwest US (Ficklin et al., 2014), the Sierra Nevada, California: (Ficklin et al., 2012), Marys River, Oregon (Mustafa et al., 2018), and Athabasca River basin in, Alberta, Canada (Du et al., 2018), showed more accurate compared to the statistical

model results proposed by Stefan & Preud'homme (Stefan & Preud'homme, 1993). Although, the model presents an explicit approach to the mass transfer process, including the main components of the mass balance of the river; the heat transfer process is simplified by the difference in temperature at the air-water interface multiplied by the flow travel time and the calibrated coefficient. Attempts to incorporate an explicit component of the energy balance into the Ficklin et al. model have included use of an equilibrium temperature approach (Du et al., 2018), and use of thermal radiation components (Mustafa et al., 2018) that are widely employed in the Heat Source model (Boyd & Kasper, 2003). These additions include an in-detail representation of the heat loss/gain components through the air-water interface such as solar radiation, atmospheric longwave radiation, back radiation, convection, and evaporation. Despite efforts to include explicit energy balance in the Ficklin et al. model, they did not consider riparian vegetation in the balance equation, which is an important factor in determining stream temperature because it blocks solar radiation from reaching the water surface (Fuller et al., 2022; Garner et al., 2017; Roth et al., 2010). Therefore, in this work we fill that gap by incorporating riparian vegetation into the energy balance equation through the equilibrium temperature approach that characterizes the heat balance at the air-water interface (Edinger et al., 1974), and we integrated it into the Ficklin et al. stream temperature model (2012) and then incorporated it into the SWAT hydroclimatological model to improve stream temperature modelling at the sub-basin level.

Riparian vegetation has been identified as an efficient strategy to control stream temperatures by blocking solar radiation from reaching streams (Chen, Carsel, et al., 1998; Roth et al., 2010; Rutherford et al., 1997). Previous studies in, for example, the US (Abbott G., 2002; Abdi et al., 2020; Chen, McCutcheon, et al., 1998), Brazil (Ishikawa et al., 2021), Europe (Johnson & Wilby, 2015; Kalny et al., 2017; Kałuza et al., 2020), Asia (Liao et al., 2014; Liu et al., 2019), New Zeeland (Rutherford et al., 1997), among other places have demonstrated the efficacy of riparian vegetation restoration in lowering stream temperatures. Riparian vegetation has also been shown to be effective in lowering silt, nutrients, and boosting biodiversity (Malkinson & Wittenberg, 2007; Poole & Berman, 2000). Furthermore, riparian vegetation also impacts hydrological and meteorological parameters. Prior research, for example, found that riparian plants like bibosoop helped to reduce wind speed and evapotranspiration in crop fields in Korean locations (Koh et al., 2010). Guenther et al. (2012) reported effects of logging on vapor pressure, wind speed, and evaporation. Rodrigues et al. (2021) also provided facts about the impact of riparian vegetation on the evaporation of reservoirs. Dugdale et al. (2018) linked riparian vegetation to changes in the flow of energy across the air-water interface and then to evaporation.

In stream temperature modelling, riparian vegetation has been represented by the shaded area over the stream generated by the canopy, either in quantity, percentage of shaded area (Li et al., 2012), or shade factor (DeWalle, 2010; Fuller et al., 2022; LeBlanc & Brown, 2000). Models for determining shading or shade factor often included hydraulic and morphological properties of the river, plant characteristics in the buffer zone, and meteorological data such as solar radiation. Complex models conducted in local scale (at specific sections of a river or short stretches of a river) even incorporated variables such as canopy shape, canopy overhang, stream bank height, canopy transmittivity, and others. Thus, these complex models also required detailed information at field level on river morphology, detailed canopy features, and in situ meteorological measurements

(Davies-Colley et al., 2009; Davies-Colley & Rutherford, 2005; Li et al., 2012). However, in large stretches of rivers where information at the field level is not feasible due to limited resources, simplified models have been employed to determine the shade factor with good enough results (Fuller et al., 2022; Marteau et al., 2022; Seyedhashemi et al., 2022; Spanjer et al., 2022). In this aspect, this study takes a simplify methodology to determinate the shade factor maintaining the more representative stream and canopy features.

The main objective of this work is to add an explicit energy-balance model that includes the shade factor of riparian vegetation into Ficklin's stream temperature model (Ficklin et al., 2012), and then integrate the improved approach into the SWAT hydrological model (Neitsch et al., 2009). After evaluating the improved stream temperature model in SWAT for Dairy McKay watershed (DMW) in Oregon, USA, this work also addressed the following related objectives: (1) Evaluate the effects of riparian vegetation on the shade factor and reductions in stream temperature, for four scenarios: full restoration along both banks of stream network, efficient restoration of riparian vegetation, current riparian conditions, and no vegetation. (2) Evaluate the reduction in the number of days above survival limits for aquatic species such as salmon, for the two scenarios of full restoration and efficient restoration of riparian vegetation in DMW.

## 2 Materials and Methods

### 2.1 Dairy McKay Watershed Case Study

The Dairy-McKay watershed (DMW) (Hydrologic Unit Code (HUC)-10: 1709001003), located in Northwestern Oregon, is part of the Tualatin sub-basin (HUC-8: 17090010). It encompasses an area of 598.3 square kilometers draining into the Tualatin River (Fig. 1). The DMW is characterized by higher elevations and varied topography of the Coast Range in the northern part and flat topography in the southern. The highest elevation corresponds to 690 masl, while the lowest one corresponds to 35 masl at the confluence with the Tualatin River. Characterized by having perennial flow, DMW is considered one of the main tributaries of the Tualatin River, which is the prominent channel within the watershed. The major area of DMW is located across Washington county (97.4%), and 1.3% across Multnomah, and the last 1.3% across Columbia County.

The DMW climate corresponds to a Mediterranean climate with the lack of rains in summer (51 mm) and mild intensity, long duration rains in winter (719 mm). DMW soils are mainly composed of fine soils such as silt and clay with abundant natural phosphate. Despite improvements in DO levels in certain streams, temperatures in a significant number of streams remain above natural values (CWL, 2019; ODA, 2018). Regarding land use, there are three main areas: the northern half area is dominated by forestry involving around 55% of the DMW, the middle part is dominated by agriculture that encompasses around 40%, and the southern part is dominated by a growing urban area by around 5%. The upstream part of the DMW is dominated by long-lived trees species such as evergreen forest and shrubland, while the downstream part is dominated by seasonal crops such as Slender Wheatgrass, and at the most downstream extent, is dominated by urban areas. Due to the predominance of fine soils, upstream areas are vulnerable to erosion and landslides phenomena (Hawksworth, 1999). In agricultural areas, water quality has been found to degrade rapidly, with higher water temperature and higher phosphorus

concentrations (CWL, 2019; Hawksworth, 1999; ODEQ, 2001). Some streams such as the West Fork Dairy Creek show lower Dissolved Oxygen (DO) levels than natural conditions, limiting aquatic life (Hennings, 2014; ODA, 2018).

## 2.2 Hydrologic Model

Hydrological processes for DMW were simulated by using the Soil and Water Assessment Tool 2012 (Neitsch et al., 2011); developed by the United States Department of Agriculture (USDA) Agricultural Research Service (ARS). The SWAT has been utilized in watershed modelling at the sub-basin level in many places across the world with outstanding results in terms of controlling flow, erosion, nitrate, and other nutrients (Abbaspour et al., 2015; Moriasi et al., 2007). The physical-based SWAT model is widely used to assess the impact of non-point sources, strategic conservation practices, conditions of soil management practices, and changes in land use in large and complex watersheds and predict their effects on flow, production of sediments and chemicals, and instream temperature (Neitsch et al., 2009). The model can simulate these hydrological processes for long periods, and at daily, monthly, and annual time steps. The study area was divided into 60 sub-basins, with areas ranging from approximately 0.41 km$^2$ to 19.4 km$^2$ (average 9.97 km$^2$), overlying as far as possible on 12-digit HUC (Hydrologic Unit Code) boundaries from DMW, which is the US hierarchical watershed classification system. For modeling purposes, each sub-basin was divided into small areas called "Hydrologic Response Units" (HRU), which are portions of areas that have unique combinations of slope topography, land use, and soil type features. Slope topography was calculated from DEM (cell size 10x10m) and classified in three ranges: 0-5%, 5-20%, and greater than 20%. The land use and soil data were retrieved from the National Land Cover Database (NLCD) and Soil Survey Geographic Database (SSURGO) in raster format with 10x10m cell size (USDA, n.d.). To eliminate small coverage areas of these features into each HRU, a threshold of 10% was considered. Hence, features with less than 10% of its HRU area were not considered as part of the combination. As a result, the SWAT model divided the DMW into 991 HRUs.

Tile drainage was considered only for agricultural areas and controlled by three parameters - depth (DDRAIN), time to drain soil to field capacity (TDRAIN), and drain tile lag time (GDRAIN), which were calibrated during flow calibration. Crop operations based on the Heat Units to maturity from the main crops (Slender-wheatgrass, red clover, winter-wheat, sweet-corn, and corn) were also considered in the watershed modeling. Stakeholders' Water Rights for irrigation purposes and Instream Water Right were also included in the watershed modeling (OWRD, n.d.). From water rights belonging to stakeholders, the allowed period to take water, the maximum volume of water allowed to take from the source, the maximum rate of water allowed to take from the source, the Points-of-Diversion (POD), and the Places of Use (POU) were considered in the model. From instream water rights, the minimum in-stream flow for irrigation diversion was considered in the model. The detailed process for including water rights in the SWAT model is available in Sect. S1 in the Supplement accompanying the article.

Precipitation and air temperature data were obtained from the PRISM Climate Group database (OSU, 2014). The dataset is available at a daily time scale and 4km spatial scale. After overlaying these data, 38 data sites were found to cover the DMW area. However, points adjacent to the basin have also been considered for modeling. Thus, 58 points were considered in the

SWAT model. Data on solar radiation, relative humidity, and wind speed were taken from the Forest Grove weather station (Long.: -123.08361, Lat.: 45.55305, Elev.: 54.9 masl) from the Columbia-Pacific Northwest Region – US Bureau Reclamation dataset (USBR, n.d.) on a daily time scale. Flow discharges and water temperature for calibration were obtained from two stations: The East Fork Dairy Creek near Meacham corner (USGS 14205400), and the Dairy Creek at RTE 8 near Hillsboro (Station ID-14206200) (USGS, n.d.). The first station lying at the sub-basin #31 outlet (see Fig. 1) was employed to calibrate the upstream DMW, while the second station lying at the sub-basin #59 outlet was employed to calibrate the downstream DMW.

## 2.3 Stream Temperature Model

Stream temperatures for DMW were simulated for four riparian vegetation scenarios. Scenario 1: simulation under current conditions of riparian vegetation. Scenario 2: simulation considering a full riparian restoration on both stream banks. The full riparian restoration contemplates the height of the trees equal to 45 m, which is the average height in the maturity stage (over 60 years) of the most common species in Oregon (Curtis et al., 1974). Scenario 3: simulation considering an efficient restoration of riparian vegetation. Here, in E-W and W-E oriented streams, the southern bank was fully restored and the northern bank was left in its present condition. The N-S and S-N oriented streams were fully restored on both banks. Scenario 4: simulation under conditions of no riparian vegetation in which both banks were parameterized in the SWAT model to have zero contribution to SF from vegetation. In DMW, 19 streams were classified as E-W and W-E oriented with azimuths in the range of 45° to 135° and 225° to 315°, and 41 streams as N-S oriented with azimuths ranging from 135° to 225°.

### 2.3.1 Stream Temperature Approach

The SWAT model by default employs a linear relationship between air temperature and stream temperature (Stefan & Preud'homme, 1993). Subsequently, Ficklin et al. (Ficklin et al., 2012) proposed an improved stream temperature model via three main components that represent the mass and energy transfer processes. The first component (Eq. 1) of the Ficklin et al. model computes the local stream temperature by mixing the snowmelt flow, groundwater, surface runoff, and lateral flow multiply by their corresponding temperatures.

$$T_{w,local} = \frac{T_{snow}\,(sub\_snow) + T_{gw}\,(sub\_gw) + \lambda\,T_{air,lag}\,(sub\_surq + sub\_latq)}{sub\_wyld} \tag{1}$$

Where $T_{w,local}$ is the local temperature; $T_{snow}$ is the snowmelt temperature; $T_{gw}$ is the groundwater temperature; $T_{air,lag}$ is the average daily air-temperature with a lag (°C); $sub\_snow$ is the snowmelt contribution to streamflow within the sub-basin; $sub\_gw$ is the groundwater contribution to streamflow within the sub-basin; $sub\_surq$ is the surface runoff contribution to streamflow within the sub-basin; $sub\_latq$ is the soil water lateral contribution to streamflow within the sub-basin; and $sub\_wyld$ is the total water yield contribution to streamflow within the sub-basin; and λ is a calibration coefficient linking the $T_{air,lag}$ and $sub\_surq$ and $sub\_latq$.

The second component (Eq. 2) of the Ficklin et al. model computes the temperature contribution of upstream sub-basin flow (tributary flows) to the streamflow within the targeted sub-basin.

$$T_{w,initial} = \frac{T_{w,upstream}\,(Q_{outlet} - sub\_wyld) + T_{w,local}\,(sub\_wyld)}{Q_{outlet}} \tag{2}$$

Where $T_{w,initial}$ is the stream temperature mixing the local temperature and the upstream streamflow temperature; $T_{w,upstream}$ is the upstream stream temperature; and $Q_{outlet}$ is the flow discharge at the outlet of the targeted sub-basin (m$^3$/d).

The third component (Eq. 3) involves terms that represent the heat transfer process and are used to adjust $T_{w,initial}$ to obtain the final stream temperature.

$$T_w = T_{w,initial} + [T_{air} - T_{w,initial}].K.TT \qquad\quad if\ T_{air} > 0 \tag{3}$$

$$T_w = T_{w,initial} + [(T_{air} + \varepsilon) - T_{w,initial}].K.TT \qquad if\ T_{air} < 0$$

Where, $T_w$ is the final stream temperature in the targeted sub-basin (°C), $T_{air}$ is the average daily air-temperature (°C), $K$ is
the bulk coefficient of heat transfer ranging from 0 to 1 (1/hr), $TT$ is the travel time of water through the sub-basin (*hr*), and $\varepsilon$ is an air temperature addition coefficient to compensate water temperatures when air-temperature is negative.

### 2.3.2 Including the Explicit Approach of Energy Balance into Ficklin et al. Model

In this research, the third component of the Ficklin et al. model (Ficklin et al., 2012)  was replaced by an explicit energy balance equation. Thus, the rate of heat transfer through the air-water interface of the stream is calculated as follows (Edinger
et al., 1974) (Eq. 4-5):

$$\frac{dT_w}{dt} = \frac{\sum H}{\rho\, C\, h} \tag{4}$$

$$\sum H = H_s + H_{at} - H_b - H_e - H_c \tag{5}$$

Where, $\Sigma H$ is the sum of heat components transferred to or released by the river (Net heat flux), $\rho$ is the water density (kg m$^{-3}$), C is the specific heat capacity (4186 J Kg$^{-1}$ °C$^{-1}$), $h$ is the water depth (m), $H_s$ is the shortwave solar radiation, $H_{at}$ is the
longwave atmospheric radiation, $H_b$ is the back radiation emitted by water to the atmosphere in longwave form, $H_e$ Is the heat loss from water to the atmosphere through evaporation, and $H_c$ is the heat gain/loss through conduction and convection. The rate of heat transfer through the air-water interface can be also represented proportional to the difference between the stream temperature and the equilibrium temperature (Eq. 6-8) (Edinger et al., 1974).

$$\frac{dT_w}{dt} = \frac{K_e\,.(T_e - T_s)}{\rho\, C\, h} \tag{6}$$

$$T_e = T_d^* + \frac{H_s}{K_e} + \frac{H_{at} - 305.5 - 4.48\,T_d^*}{K_e} \tag{7}$$

$$K_e = 4.48 + 0.05\,T_s + (\beta + 0.47)\,.f(W) \tag{8}$$

Where, $T_e$ is the equilibrium temperature defined as the hypothetical water temperature at which the net heat flux is zero, $T_d^*$ is the modified dew-point temperature. Brady, Graves, and Geyer (Brady et al., 1969) have found negligible loss in accuracy

when the modified dew-point temperature is assumed to equal to the original dew-point $T_d^* \approx T_d$; however, in this study, the second term will be represented by a constant value (Eq. 9) that will be calibrated.

$$T_d^* \approx T_d + c_o \tag{9}$$

For air temperatures ranging from 0 to 30 °C, the relationship between the air and dew-point temperature is nearly linear. Considering that more than 97% of the DMW air temperature over a year is within this range (0-30 °C), we can assume a linear relationship between the air and the modified dew-point temperature (Parish & Putnam, 1977) (Eq.10).

$$T_d^* \approx C_1 T_a + C_2 \tag{10}$$

Where $C_1$ and $C_2$ are constants to be calibrated in the model. However, since the dew-point is always lower than or equal to the air temperature, the coefficients were constrained to get $T_d < T_a$.

The short-wave radiation reaching the water surface is equal to the difference between the potential solar radiation and the radiation blocked by barriers such as topography and riparian vegetation. This difference can also be expressed in terms of the shadow that the barriers generate over the streams as a factor (Abdi et al., 2020; Boyd & Kasper, 2003) (Eq. 11).

$$H_s = 0.97 \, H_{day}(1 - SF) \tag{11}$$

Where: $H_{day}$ is the incident total solar radiation per day (MJ/m$^2$.day), $SF$ is the shade factor.

The longwave radiation ($H_{at}$) emitted by the atmosphere is computed by the Stefan-Boltzmann law (Hebert et al., 2011; Kim & Chapra, 1997; Morin & Couillard, 1990).

## 2.4 Shade Factor Approach

The shade factor was calculated as the portion of solar radiation blocked by the topography and riparian vegetation divided by the potential solar radiation that would reach the stream surface (Boyd & Kasper, 2003). Thus, the shade factor varied from 0 (when no solar radiation is blocked) to 1 (when all the potential solar radiation heading toward the stream is blocked). The amount of radiation blocked by the barriers depended on the size and proximity of trees, topographic angle, solar azimuth, solar angle, stream width, stream azimuth, stream coordinates, the percentage of radiation solar that penetrates the canopy, and date/time. Thus, the shade factor was different for each stream, each day within the year, and each instant within the day. This calculation process was performed in the Python environment (available at https://github.com/noayarae/SF_model.git) and then the results were input into the SWAT hydrological model.

The existing vegetation Height (EVH) data was obtained from the Land-fire Program (LP) database (LANDFIRE, 2019) in raster format with 10x10m cell size. The average height in a 30 m buffer was obtained. The proximity of trees was assumed constant and equal to 5.0 m which is approximately equal to the average crown radius of the major tree species of Oregon at maturity (Bechtold, 2003; Temesgen, H., Hann, D.W., & Monleon, 2007). Since forests and riparian vegetation in DMW were mostly evergreens (LANDFIRE, 2019) that keep their leaves year-round and maintain a nearly constant high average leaf area index throughout the year (Ishikawa et al., 2021; Thomas & Winner, 2000), the shade factor did not consider seasonal changes in the leaf area index of riparian vegetation. For the scenarios of full and efficient riparian restoration, we also assume that this

type of vegetation will be planted. However, in rivers buffered by other types of vegetation, seasonal defoliation may be relevant to consider

The topographic angle, measured between the middle of the stream and the highest topographic feature in a radius of 50 km, was calculated from the DEM in the GIS environment for each river and each solar azimuth. When the topographic angle was greater than the riparian vegetation angle, the blocked solar radiation was assigned to the topography.

Thus, the SF has been calculated for each day of the temperature simulation period and for each DMW stream. To simplify, assumptions about the geometry has been considered (Li et al., 2012). The SF estimate did not consider, for example, the geometry of the trees or the density of the riparian vegetation. The detailed process for calculating the shade factor is available in Sect. S2 in the Supplement accompanying the article.

## 2.5 Model Calibration Setup

Flow and stream temperature calibration processes were performed at the East Fork Dairy Creek and at Dairy Creek on RTE 8 station, which are outlets of the sub-basin #31 and sub-basin #59, respectively. The SWAT-CUP tool was used to calibrate the flow by changing seventeen parameters (Detail of the calibrated parameters are available in Sect. S3 in the Supplement). Flow calibration was carried out in sub-basin #31 (upstream of the DMW) from 1/1/2006 to 12/31/2018, and in sub-basin #59 (downstream of the DMW) from 5/5/2011 to 31/12/2018. The calibrated parameters in sub-basin # 31were extended to the other upstream sub-basins with similar physical characteristics to the sub-basin #31, while the calibrated parameters in sub-basin # 59 (downstream) were extended to downstream sub-basins. On the other hand, the proposed stream temperature model was calibrated at the outlet of sub-basin #31 from 2/16/2012 to 12/31/2008, and at the outlet of sub-basin #59 from y 1/1/2006 to 3/5/2012. The calibration was accomplished by an iterative procedure that was systematized in Python code following the steps shown in table S4 and S5 in section S6 in the Supplement. The Python code to iteratively run SWAT, the input data, required SWAT files, and the modified SWAT model (in Fortran) may be found in the Zenodo repository at https://doi.org/10.5281/zenodo.6301709 (Noa-Yarasca, 2022). Calibration was carried out by changing four parameters ($\lambda$, tair_lag, parameters from the predeterminate Ficklin et.al. model, and $C_1$ and $C_2$, coefficients introduced in this study). Following the Latin Hypercube Sampling criterion (Iman, 2008), 2000 sample sets of the four coefficients ($\lambda$, tair_lag, $C_1$, and $C_2$) were generated and iteratively evaluated to find the optimal values of the parameters.

## 2.6 Model Calibration Evaluation

The model's efficiency was assessed using the Nash Sutcliffe efficiency criteria (NSE), which is given by the equation below.

$$NSE = 1 - \frac{\sum_{i=1}^{n}(O_i - S_i)^2}{\sum_{i=1}^{n}(O_i - O_{avg})^2}$$

Where $O_i$ is the observed value at time $i$, $S_i$ is the modeled value at time $i$, $O_{avg}$ is the mean of observed values. NSE values range from $-\infty$ to 1, with 1 indicating a perfect model with zero prediction error, NSE = 0 indicating a model with predictive

power equal to the mean of observed values, and negative values indicating a very severe model error with prediction worse than the mean of observed data. Previous research has classified models with NSE values less than 0.5 as unsatisfactory, models with values more than 0.65 as good, and models with values greater than 0.75 as very good (Moriasi et al., 2007). In addition, the average tendency of the simulated values to be greater or lower than their observed values were measured by percent bias (PBIAS), given by

$$PBIAS = 100 \, \frac{\sum_{i=1}^{n}(O_i - S_i)}{\sum_{i=1}^{n} O_i}$$

Where $O_i$ is the observed value at time $i$, $S_i$ is the modeled value at time $i$. PBIAS has an optimum value of 0, with values close to zero suggesting accurate model simulation. Positive values imply overestimation bias, whereas negative values suggest underestimating bias in the model.

Moreover, model error was performed using the mean absolute error (MAE), given by

$$MAE = \frac{1}{n}\sum_{i=1}^{n} |S_i - O_i|$$

This is an arithmetic average of the absolute errors between paired observed and simulated values. The MAE ranges from 0 to ∞. Given that it is a negatively oriented score, models with low MAE are preferable, with MAE = 0 being the ideal model.

## 3 Results and Discussion

### 3.1 Flow Calibration

For flow, the calibrated model achieved a Nash Sutcliffe Efficiency (NSE) of 0.74 for sub-basin #31 and 0.86 for sub-basin #59. The PBIAS values obtained were 8.9% for sub-basin #31 and 6.4% for sub-basin #59. These efficiency values are consistent with calibrations performed for other watersheds (Arnold et al., 2012; Moriasi et al., 2007), in which the NSE for the flow calibration ranged between 0.58 and 0.98 and the PBIAS was less than 10%. Figure 2a-b shows the performance of the calibrated model.

### 3.2 Stream Temperature Calibration

### 3.2.1 Shade Factor

The shade factor in DMW streams varied both temporally and spatially. Temporally on average, the shade factor in winter was found to be greater than in summer. Spatially, the shade factor ranged from 0.001 in streams with very little riparian vegetation to 0.91 in streams with existing vegetation with tall trees. Note that values of shade factor for each stream for the existing vegetation and other scenarios have also been graphed are discussed more in detail in section 3.3.1. In addition to existing vegetation and topography, the temporal variation of the SF was driven by variation in solar declination and solar azimuth during the year, while the spatial variation SF was driven primarily by stream orientation. Thus, the contribution of riparian

vegetation and topography in blocking the solar radiation, and therefore in the shade factor, was mainly conditioned by the stream orientation (varying spatially), solar declination, and solar azimuth (varying temporally).

Overall, the contribution of topography to the shade factor was found to be small compared to the contribution of riparian vegetation. For example, considering that the SF goes from 0 to 1, the topography contribution was found to be from 0.001 to 0.08 while the riparian contribution was found to be from 0.01 to 0.87. The contribution of topography to SF was found to be

even lesser in downstream streams than in upstream streams. For example, the average contribution of topography in the SF in upstream streams was 0.04 while in downstream streams it was 0.004. This means that the amount of solar radiation blocked by the topography was considerably less than the amount blocked by the riparian vegetation in this watershed; however, in rivers surrounded by high ridges, the topographic contribution may be more relevant.

Regarding riparian vegetation, because the DMW is located in the Northern Hemisphere, solar declination greatly favored to

the southern bank riparian vegetation to shade EW and WE oriented streams rather than the northern side. Therefore, the southern bank contribution to the SF was significantly greater than that of the northern bank in EW and WE oriented streams. However, in streams located in the Southern Hemisphere, this contribution would be inverse. In NS and SN oriented rivers, the contribution of riparian vegetation from the western and eastern banks to the SF were similar over the year.

### 3.2.2 Calibration

The values of the four calibrated coefficients ($\lambda$, tair_lag, $C_1$, and $C_2$) driving the modified stream temperature model were 0.88, 5, 0.67, and 1.16 for sub-basin #31 and 1.06, 6, 0.74, and 1.17 for sub-basin #59, respectively. The Nash Sutcliffe Efficiency (NSE) values for sub-basin # 31 and # 59 were 0.74 and 0.82, respectively. These two NSE values are considered as good fit and very good fit (Moriasi et al., 2007),  respectively, and are consistent with successful calibrations reported in other studies ranging from 0.70 to 0.89 (Du et al., 2018; Ficklin et al., 2012; Mustafa et al., 2018). Figure 3a-b shows the

performance of the calibrated model. Although the calibration achieved encouraging evaluation coefficients, the gap between observed and simulated values during the winter at the upstream control point (sub-basin #31) is notable compared to other periods. In this period and zone, the stream temperature may be influenced by additional factors or variables that have not been considered in this study. These factors can be, for example, canopy density changes in winter, hyporheic flow, heat from winter precipitation, bottom friction heat in winter flows, and others. Future research is recommended to take these aspects into

account.

The stream temperature calibration using the modified Ficklin et al. model highly outperformed stream temperatures computed by using the Stefan's equation (Linear model currently used as the default approach in SWAT). On the other hand, the accuracy of the modified model was found to be fairly similar (within ±0.05 NSE of each other) to the original Ficklin et al. model (Table 1). Residual values of stream temperature simulated by the linear model, calibrated by the original and the modified

Ficklin et al. model for Sub-basin #31, and Sub-basin #59 are also shown in Fig. 4a-b.

### 3.3 Evaluating the Effects of Riparian Vegetation on Stream Temperature

Data of existing vegetation of the main DMW streams show non-forested banks in 45.3%, partially forested in 42.5%, and only 12.2% of high forested banks, indicating that there is still a significant amount of buffer zone to reforest and an important amount of solar radiation heading to the streams to be blocked. However, the restoration of all potential vegetation can become
a costly alternative as financial resources are often limited (Minnesota Board of Water and Soil Resources, 2009). Hence, the optimization of potential riparian then results to be an effective option to find the most favorable riparian without sacrificing the goal of stream temperature reduction.

#### 3.3.1 Effects of Riparian Vegetation on the Shade-Factor

The full (Scenario 2) and efficient (Scenario 3) riparian restoration resulted in increases in the Shade Factor (SF) with respect
to the existing riparian vegetation (Scenario 1), in all the 60 DMW streams (Fig. 5). In streams with no forested and partially forested banks, substantial SF increases were obtained. For example, the SF of Stream #20 in sub-basin #20 increased from 0.002 (current SF) to an average of 0.93 under full reforestation and to 0.86 under efficient reforestation. In areas forested with relative tall trees, minor increases in SF have been obtained. For example, the SF of Stream #1 increased from 0.82 (current SF) to 0.89 (in full and efficient riparian reforestation).
The contribution of riparian vegetation in the SF varied according to the stream orientation (azimuth) and the stream bank. To illustrate, in the stream #20 (with azimuth 107.5° - near WE orientation), in full riparian restoration, the southern bank contributed 92.2% in the SF increase, while the northern bank contributed in only 7.7%, and the topography contribution was 0.1%. In efficient riparian restoration (scenario 3), the northern bank was not considered to be reforested; therefore, the SF increase is only due to the southern bank reforestation.
Overall, due to DMW's location in the Northern Hemisphere, in streams with a dominant E-W and W-E orientation, the contribution to SF from the southern side riparian vegetation was greater than that from the northern side. In streams with a dominant N-S and S-N orientation, the contribution to SF from the eastern and western side riparian vegetation were similar. Details showing the contribution of stream banks to the SF increase are available in Sect. S4 in the Supplement.

#### 3.3.2 Reduction of Mean Stream Temperature

The full and efficient riparian restoration resulted in stream temperature reductions with respect to the existing riparian vegetation, in all the 60 DMW streams. In both riparian restoration scenarios, average annual temperature reductions in stream segments ranged from 0.02 to 3.17 °C, compared to current conditions (Scenario 1). Despite the same ranges, the mean reductions in stream temperatures for full riparian restoration was 0.91 °C (SD = 0.69 °C) while for efficient restoration it was 0.86 °C (SD = 0.67 °C). In summer period, these reductions ranged between 0.03 and 5.21 °C, with a mean of 1.40 °C (SD =
1.17 °C) for full riparian restoration and 1.31 °C (SD = 1.13 °C) for efficient riparian restoration (Fig. 6a-b). Reductions in

stream temperature were found to be directly proportional to increases in shading factor. Thus, streams with substantial increases in SF also showed substantial reductions in stream temperatures.

As in the SF analysis, in streams with a dominant E-W and W-E orientation, the contribution to stream temperature reduction of riparian vegetation on the southern side was greater than that on the northern side. In N-S and S-N oriented streams, both the eastern and western banks contributed to the stream temperature reduction in a similar way. Details showing the contribution of stream banks to the stream temperature reduction are available in Sect. S5 in the Supplement accompanying the article. Stream temperature reductions for full and efficient riparian restoration were quite similar. This implies that a strategic allocation of riparian vegetation can achieve levels of stream temperature reduction as well as a full restoration. This finding is consistent with previous studies seeking strategic placement of riparian vegetation to achieve the greatest reduction in water temperature. DeWalle (2010), for example, discovered that during summer solstice, south bank riparian vegetation in E-W streams produced 70% of total daily shade compared to 30% of north bank on a 40°N stream, while in N-S streams shading from both banks were equivalents. Similarly, Garner et al. (2017), reported that planting on the southernmost bank of Northern Hemisphere streams flowing E-W, NE-SW, or NW-SE, and vice versa, would result in optimal planting targeted at cooling stream water due to its greater contribution in shadowing compared to the northern bank. Likewise, Jackson et al. (2021), found that in E-W/W-E oriented rivers, the contribution of the north bank riparian vegetation was negligible when compared to the south bank. Thus, tree planting on the north side may be unnecessary for stream temperature control. In N-S/E-N oriented streams, the riparian vegetation on both sides had the same shading effect on streams.

### 3.3.3 Reduction of the Number of Days with 7-Day Average Maximum Stream Temperature greater than 18 °C

The 7-day average maximum (7dAM), calculated by averaging the daily maximum stream temperatures for 7 consecutive days, is the biologically based numeric temperature criteria to characterize the beneficial use of freshwater (ODEQ, 2008). To evaluate the model performance in relation to biological criteria, we used the numeric temperature criteria corresponding to salmon & trout rearing migration, which establish that the 7dAM do not exceed 18°C (ODEQ, 2008). In full riparian restoration, the reduction of the number of days exceeding 18°C over the year in average varied from 0 to 58.5 days ($M = 17.1$) (Fig. 7a) and over the summer from 0 to 33 days ($M = 11.4$) (Fig. 7b). The lowest reduction was observed in Stream #25, while the greatest reduction was observed in Stream #49. In efficient riparian restoration, the reduction of the number of days exceeding 18 °C over the year and over the 60 DMW streams, varied from 0 to 58.5 days ($M = 16.2$), and over the summer varied from 0 to 29.4 days ($M = 10.6$ °C), on average. These reductions were consistent with the increase of the SF. In the sub-basins with higher SF increases, the reduction of the number of days exceeding 18 °C were also found to be higher.

Previous studies have also obtained positive relationships between increased riparian vegetation and reduced stream temperature using various metrics (Abbott G., 2002; Garner et al., 2017; Kalny et al., 2017; Parkyn, 2004; Wondzell et al., 2019). However, given that future climate change scenarios foresee prolonged hot days that would affect aquatic life (Brander, 2007), this work presents the reduction of days with 7dAM that exceed 18°C, which could be a more practical value/metric

for experts and non-experts. The reduction in the number of days with 7dAM indicates encouraging findings for DMW; nevertheless, it was not able to compare with earlier research since they directly concentrate on temperature reduction under various conditions.

### 3.3.4 Cost of Riparian Restoration

Considering the vegetation density data in a 30-meter buffer zone, approximately 1900.7 acres of no forested areas and 1,770.6 acres of partially forested areas could be restored in the DMW. Riparian restoration costs vary according to factors such as location and technology used. In 2010, the ODEQ estimated the average cost of restoring riparian vegetation in rural areas in 4,695 USD per acre (ODEQ, 2010). Based on the cost value estimated for 2010, the full riparian restoration of DMW streams could cost 12.27 million USD, while an efficient riparian restoration could cost 10.51 million USD. Therefore, the efficient riparian restoration could be 14.4% cheaper in cost than the full restoration, while in terms of benefits of reducing stream temperature and reducing the number of days exceeding 18 °C, efficient restoration would have achieved more or less similar results to full restoration. Using the reduction in the number of days exceeding 18 °C as a metric for benefit of riparian restoration, the Benefit-Cost Ratio (BCR) was determined as an indicator of investment efficiency in the 60 DMW streams (Fig. 8). The BCR values show that headwater streams obtain greater benefits from riparian vegetation restoration per investment cost than downstream streams in both full restoration (Scenario 2) and efficient restoration (Scenario 3) (Fig. 9a-b). The detailed process for calculating the cost of riparian restoration/reforestation is available in Sect. S7 in the Supplement accompanying the article.

If riparian vegetation could be planted along the entire length of the river, the main expected impact would be a reduction in nutrients, sediments, overflows, and stream temperature in various measures, as well as changes in certain sub - processes of the water cycle in the stream environment such as transpiration and aquifer recharge, among others. Other expected consequences include the loss of some primary food producers, which may affect the food chain near the river. The findings of effective riparian vegetation restoration in this work are focussed on a single goal: stream temperature. These results may vary in a multi-objective assessment of riparian vegetation restoration. Further work is encouraged to assess and evaluate the implementation of multi-target riparian vegetation.

### 3.3.5 Evaluating additional effects of riparian vegetation for optimal restoration (future research)

In addition to the positive impacts of riparian vegetation on stream temperature reduction revealed here and earlier research (Abbott G., 2002; DeWalle, 2010; Garner et al., 2017; Kalny et al., 2017; Roth et al., 2010; Sahatjian, 2013), other impacts should not be overlooked when evaluating the implementation of buffer vegetation. Riparian vegetation has also been linked to other services such as reducing nutrients in streams caused by agricultural and livestock activity (Groh et al., 2020; Lutz et al., 2020), controlling soil erosion and bank stability (Dickey et al., 2021), and controlling storm runoff by slowing down water contribution to streams, absorbing rainwater, and allowing groundwater recharge, among others (Hawes & Smith, 2005). While water temperature regulation is based on the canopy's capacity to block solar radiation, other riparian-vegetation services are

linked to plant functional features such as root absorption capability, root density, and root depth. The efficient restoration of riparian vegetation reported in this work does not necessarily imply effective restoration for other purposes (nutrient reduction, flow, and erosion control), since these other services are related not only to the canopy but also to other plant functional properties(Hawes & Smith, 2005; Malkinson & Wittenberg, 2007).

A riparian buffer consisting of a mix of trees, shrubs, and grasses is much more efficient in removing a broad range of
contaminants than a riparian buffer consisting primarily of trees. This is because grasses' shallow and dense roots are excellent in slowing overland flow and trapping sediments, whereas tree roots are good at absorbing nutrients from groundwater, stabilizing banks, and regulating streamflow (Hawes & Smith, 2005). Furthermore, trees provide shade to cool the water, habitat for birds and other wild critters, and falling leaves and branches provide a source of food for wildlife and aquatic animals. Thus, grasses and shrubs can provide services that forests cannot (Parkyn, 2004).

On the other side, fully riparian vegetation restoration may greatly increase transpiration on hot days, resulting in greater water extraction from rivers by plants, which may be temporarily detrimental to sensitive aquatic species (Garner et al., 2017; Hernandez-Santana et al., 2011). Furthermore, heavy shade could affect the population of primary food producers such as periphyton and grazing snails, which are important oxygen providers for secondary consumers, water quality regulators, home to tiny creatures, and soil moisture reservoirs (Hill et al., 1995; National Park Services, 2020; Schiller et al., 2007).

If riparian vegetation could be planted along the entire length of the river, the main expected impact would be a reduction in nutrients, sediments, overflows, and stream temperature in various measures, as well as changes in certain sub - processes of the water cycle in the river environment such as transpiration and aquifer recharge, among others. Other expected consequences include the loss of some primary food producers, which may affect the food chain near the river. The findings of effective riparian vegetation restoration in this work are focussed on a single goal: stream temperature. These results may vary in a
multi-objective assessment of riparian vegetation restoration. Further work is encouraged to assess and evaluate the implementation of multi-target riparian vegetation.

### 3.3.6 Model limitations and uncertainties

Although this study achieved encouraging findings in terms of stream temperature decrease through the implementation/restoration of riparian vegetation, it should be noted that these results may vary due to the uncertainty that the
proposed model entails. The stream temperature model outcomes are subject to uncertainties arising from sources, such as input data, model structure, and model parameters.

The input data included hydrometeorological observations (air temperature, wind speed, solar radiation, and humidity), surface (land cover, vegetation height), and subsurface watershed features (groundwater properties), among others that were mainly obtained from open-source repositories on the web. Land cover and riparian vegetation (important component in this work),
for example, are derived from preprocessed satellite measurements. Despite significant improvements in remote sensing measurement techniques, errors and uncertainties remain when adapting this data to the model. For example, raster vegetation data with a cell size of 10x10m weights land cover features that are not necessarily uniform within the cell area. Furthermore,

remote sensing reads the vegetation at different time intervals and interpolates/extrapolates or weights it for the entire year or certain seasons of the year; yet, the vegetation is dynamic at all times. In reference to solar radiation, for example, given the limited number of measurement locations, the Forest Grove station's records for solar radiation were extended over the whole DMW. Despite the short range of latitudes and longitudes, which could assume uniform solar radiation in the DMW, variables like cloud cover would make this variable changeable in space. Furthermore, to calculate the daily shade factor by integrating instantaneous shade factors (0.01 hrs.), the available daily solar radiation was downscaled using models available in the literature, which also carried uncertainty.

The temperature model also included several parameters, coefficients, and constants that, despite our meticulous and judicious selection and assumption of these parameters, the certainty of many of them could not be verified directly but had to be assumed based on the similarity of our case of study with the cases for which they were obtained. This inevitably generated another source of uncertainty in the proposed model. Among these parameters are, for example, the coefficients of the various thermal sources in the Edinger's (1974) equilibrium temperature equation, and the coefficients of the Ficklin equation, which, while some of them were calibrated, could not be homogenous throughout the DMW. In terms of parameter calibration, this was performed during the relatively short time of data available for flow and stream temperature; however, a calibration over a longer period might modify the calibrated parameter values and hence the model outcomes.

With respect to structure uncertainty, the equilibrium temperature model assumes simplifications to facilitate calculation and overcome some limited resources. Among these is the linearization of the quartic energy balance equation, which in its original form is challenging to manipulate and incorporate into the hydrological model. Furthermore, based on Brady et al. (Brady et al., 1969) and Lawrence's (Lawrence, 2005) proposal, the relationship between air and dew-point temperature reduces to a linear equation, indicating that errors with this assumption may be ignored.

Finally, as with any modeling process, this work attempted to gather and preprocess high-quality input data, assumed parameters using physical criteria, and simplified certain equations to facilitate stream temperature modeling and integrate it into the hydrological model, but these efforts inevitably introduced uncertainties. Findings of this study are limited to the ranges used here, as well as the DMW characteristics. Future research should continue exploring larger variable and parameter ranges to generalize this approach. This work can be enhanced by re-adjusting the calibrated parameters and lowering the uncertainty to the extent that measurements of the input variables are available across longer and more spatially dense periods in the basin.

**4 Conclusions**

This study presented and evaluated a stream temperature modeling approach for the Soil and Water Assessment Tool that integrates an explicit energy balance model to simulate the heat transfer process influenced by riparian shading. The energy balance equation incorporated into the hydrometeorological model included the three main sources of energy (shortwave radiation, longwave radiation, evaporation, and conduction). The riparian vegetation was included through the shade factor in the shortwave radiation equation. An approach for calculating shading factor was proposed and used to evaluate the effects of

riparian shading on blocking the solar radiation and reducing the stream temperatures. The capability of the original Ficklin et al. model was improved by enabling mechanisms for capturing the cumulative effects of riparian vegetation shading on the stream temperature within the watershed. Unlike other models, this approach shows the stream temperature simulation at sub-basin scales considering detailed processes of both heat and mass transfer.

The topographic influence was also assessed, though its influence on the shade factor and the temperature of the streams were found to be very small at the testbed site. Because of DMW's location in the Northern Hemisphere, solar declination angles during the year were mostly favorable for southside riparian vegetation to shade E-W and W-E oriented streams more than northside riparian vegetation. Therefore, the contribution of the southside riparian vegetation to the increase in SF and the reduction in stream temperature were more relevant than the northside riparian vegetation in EW and WE oriented streams.

Conversely, in SN and NS oriented streams, shading and contribution of eastern and western banks were similar. Simulations showed that full riparian restoration would reduce the stream temperature on average by 0.91 °C (SD = 0.69 °C) and efficient restoration; by 0.86 °C (SD = 0.67 °C). These reductions were observed mostly in summer than in any other season. In reducing the number of days that exceed 18 °C (biological temperature threshold of aquatic species), full riparian restoration could achieve a reduction in the range of 0 to 58.5 days in the year with an average of 17.1 days. A similar range of reduction could

be achieved with the efficient restoration but with a mean of 16.2 days. Lastly, the efficient riparian restoration could be 14.4% cheaper than the full riparian restoration.

The SWAT model that computes the effects of land management practices on water flow, nutrients, and stream temperature has been successfully applied in several watersheds in the US and around the world. Similarly, the effectiveness of riparian vegetation in reducing stream temperature was demonstrated in several rivers. Therefore, the application of the improved

stream temperature model could be replicated in other regions with characteristics similar to the DMW. However, it is important to note that while the proposed temperature model improves SWAT's ability to simulate riparian buffers as a conservation practice for stream temperature management, this model did not consider the shape of the trees, nor the density of the riparian vegetation. Other considerations such as hyporheic exchange processes, frictional heat exchange, stream geometry that influence stream energy balance were also not incorporated. All of these are recommended directions for future

work.

**Acknowledgments**

This study has been supported by National Oceanic and Atmospheric Administration (Award ID: NA16OAR4320152). We would also like to thank all of our collaborators from the different agencies and institutions: Dr. Daniel Sobota at Oregon

Department of Environmental Quality, Dr. Adriana D. Piemonti, Dr. Scott Mansell, and Dr. Ting Lu at Clean Water Services, Hillsboro OR, and Dr. Sammy Rivera at Oregon State University for the input data and insights provided during this research.

### Data Availability

The data employed in this study is available as: Efrain Noa-Yarasca. (2022). Data on An Improved Model of Shade-affected Stream Temperature in Soil & Water Assessment Tool. https://doi.org/10.5281/zenodo.6301709. These data include land cover, soil type, water rights, weather (precipitation, temperature, solar radiation, humidity, and wind speed), flow and stream temperature, modifications to the SWAT rev681 program, and the calibrated DMW SWAT model.

### Supplement

The supplement related to this study is divided into seven sections that accompany the article.

### Author contributions.

The paper was written by ENY with contributions from all co-authors. ENY collected the data. All authors designed the study. ENY conducted the modelling and analysis. All authors discussed the results and gave critical feedback on the paper.

### Competing interests

The authors declare that they have no conflict of interest.

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

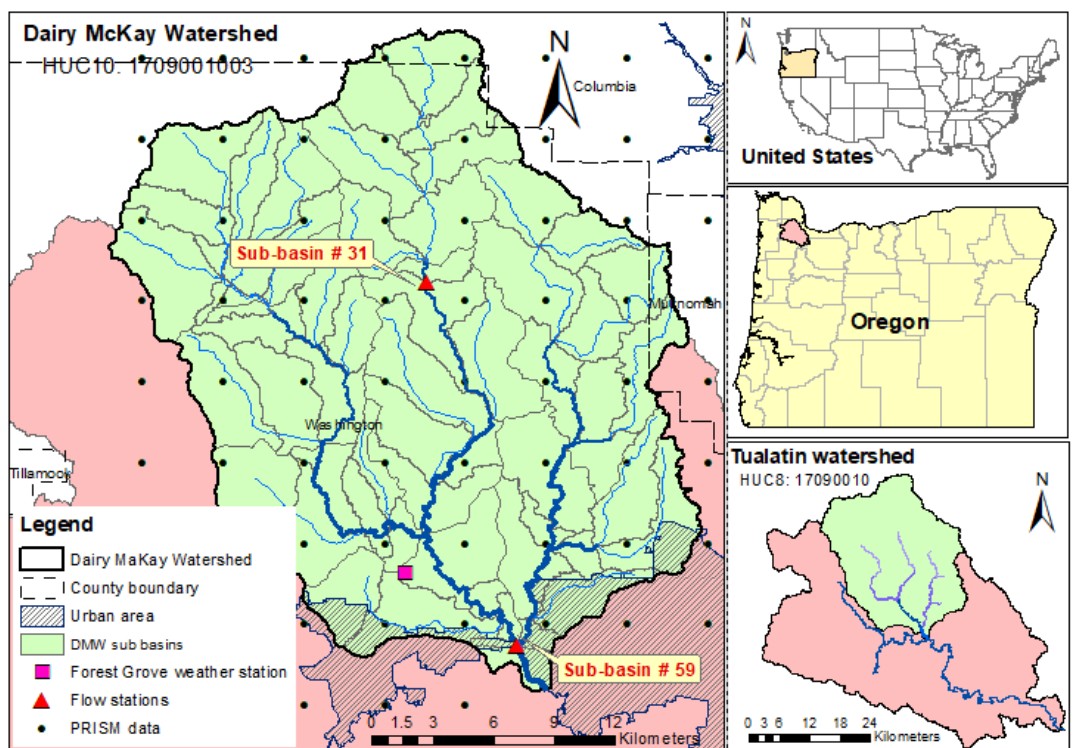

**Figure 1: Left, Streams, sub-watersheds, and political boundaries of the Dairy McKay Watershed (DMW) (HUC10-1719001003). Top right, location of DMW in the Tualatin River basin, and Bottom right, location of the Tualatin River basin in North-western Oregon, USA.**


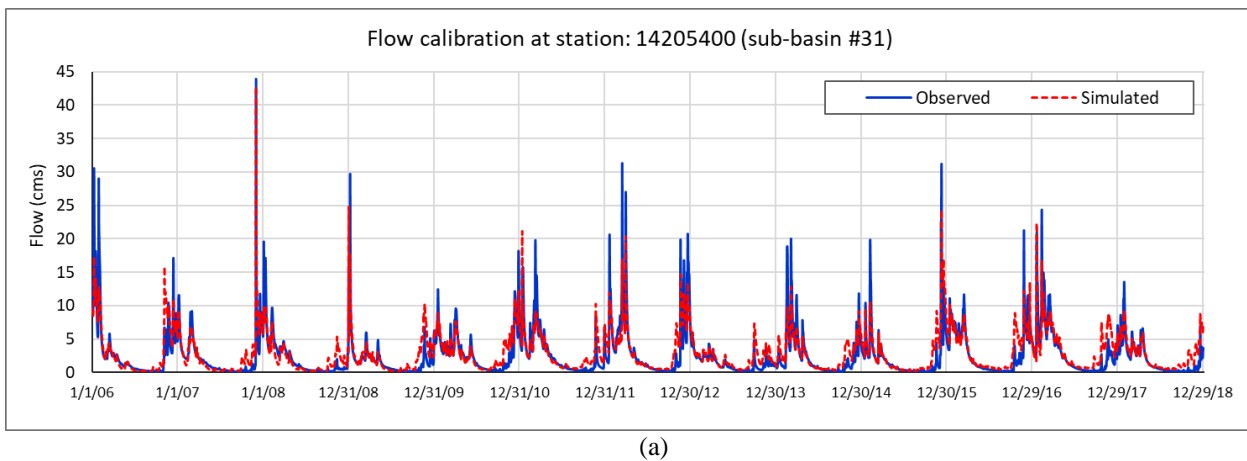

(a)

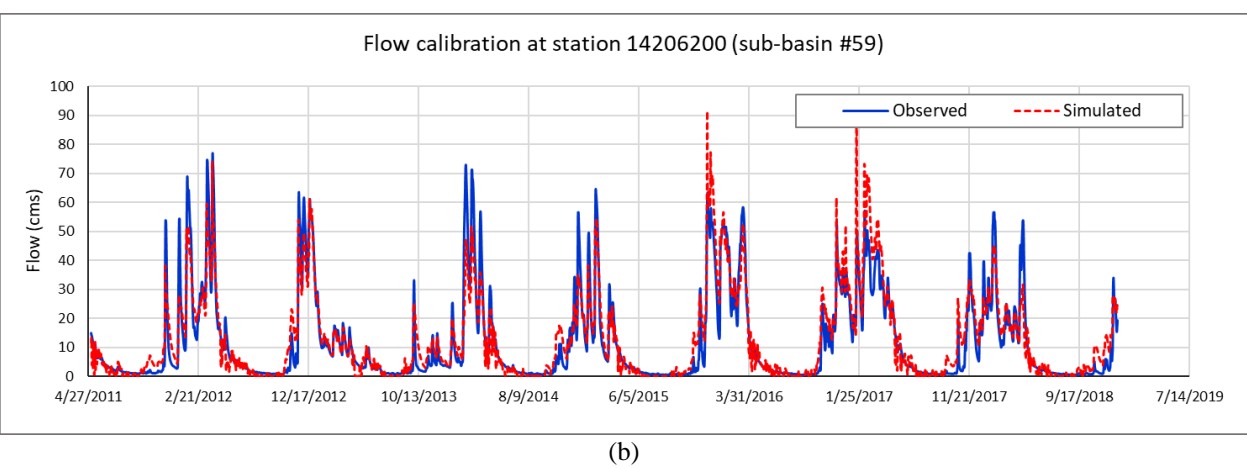

(b)

**Figure 2. (a) Flow calibration at station USGS-14205400 (outlet of sub-basin #31), and (b) station 14206200 (outlet of sub-basin #59) (bottom).**

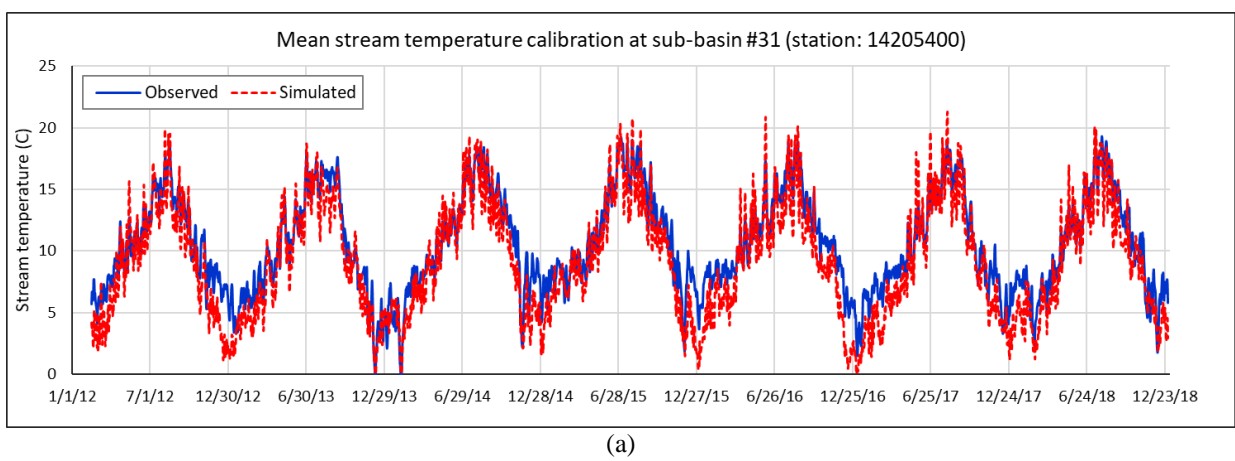


(a)

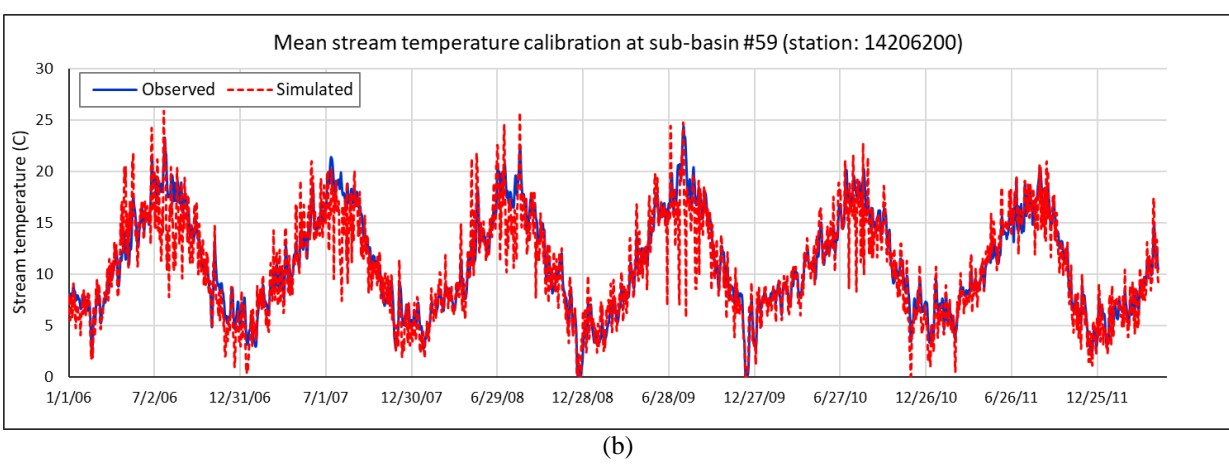

(b)

**Figure 3. (a) Mean stream temperature calibration at USGS-14205400 station (outlet of sub-basin #31), and (b) station 14206200 (outlet of sub-basin #59) (Bottom).**


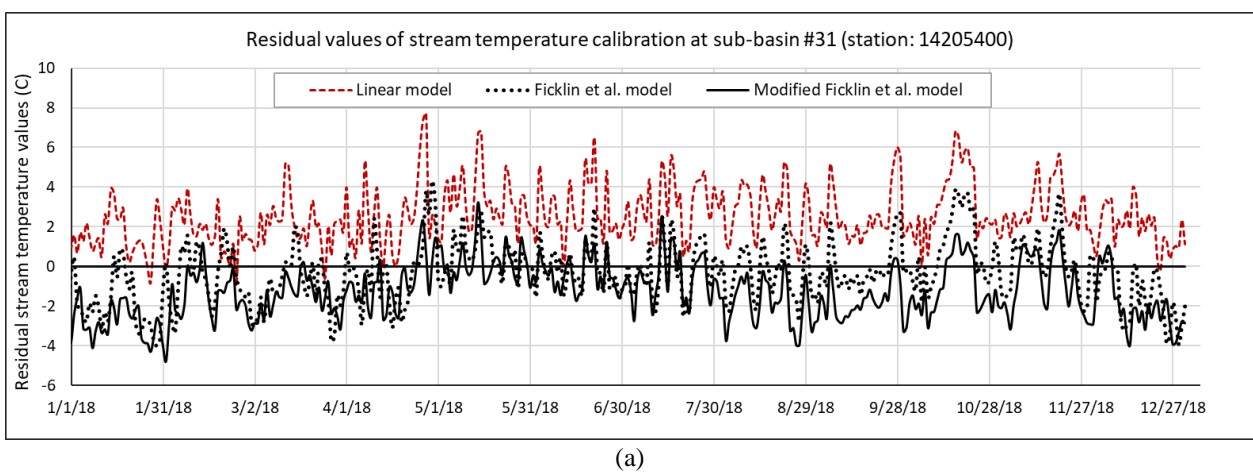

(a)

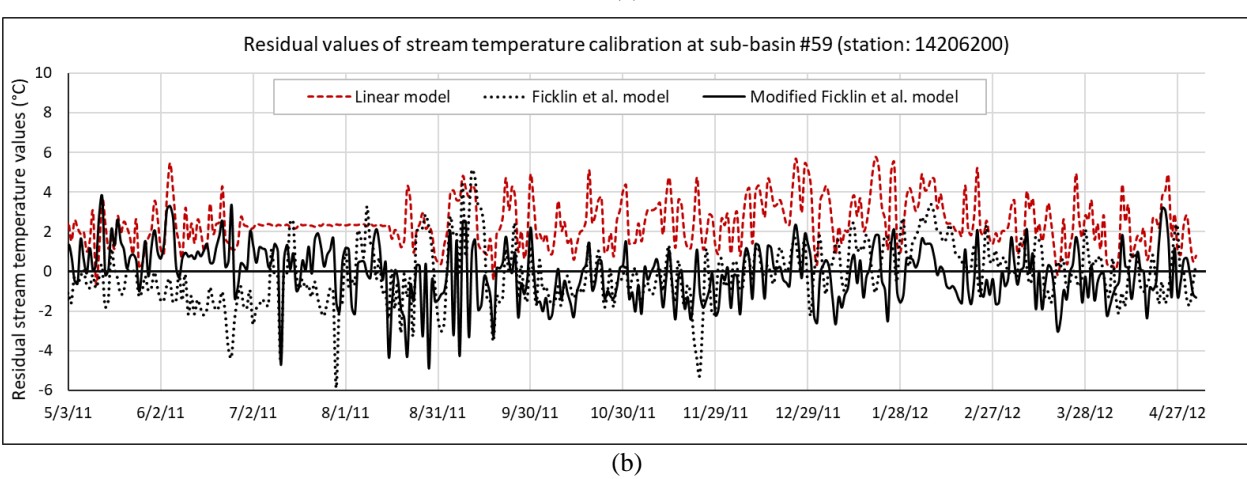

(b)

**Figure 4. Residual values of stream temperature simulated by the linear model, calibrated by the original and the modified Ficklin et al. model for (a) Sub-basin #31, and (b) Sub-basin #59 for the last year of the calibrated period.**



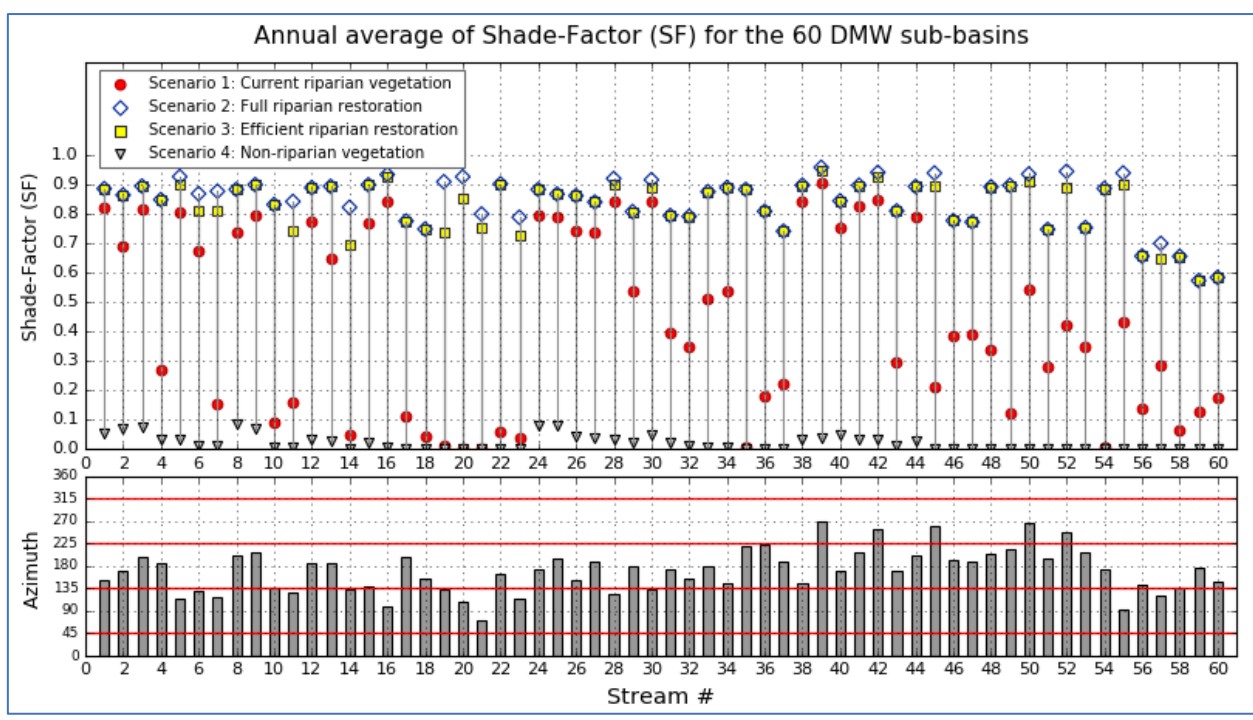

**Figure 5. Annual average of Shade Factors (SF) for the 60 DMW streams for scenarios 1, 2, 3 and 4.**

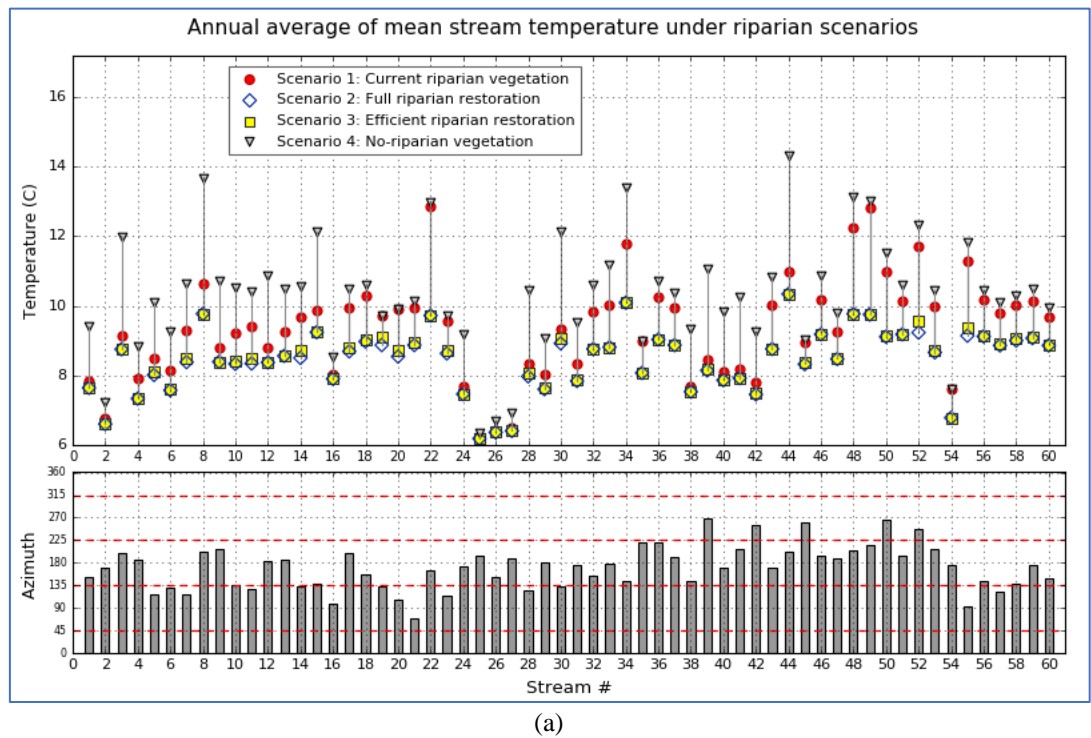

(a)

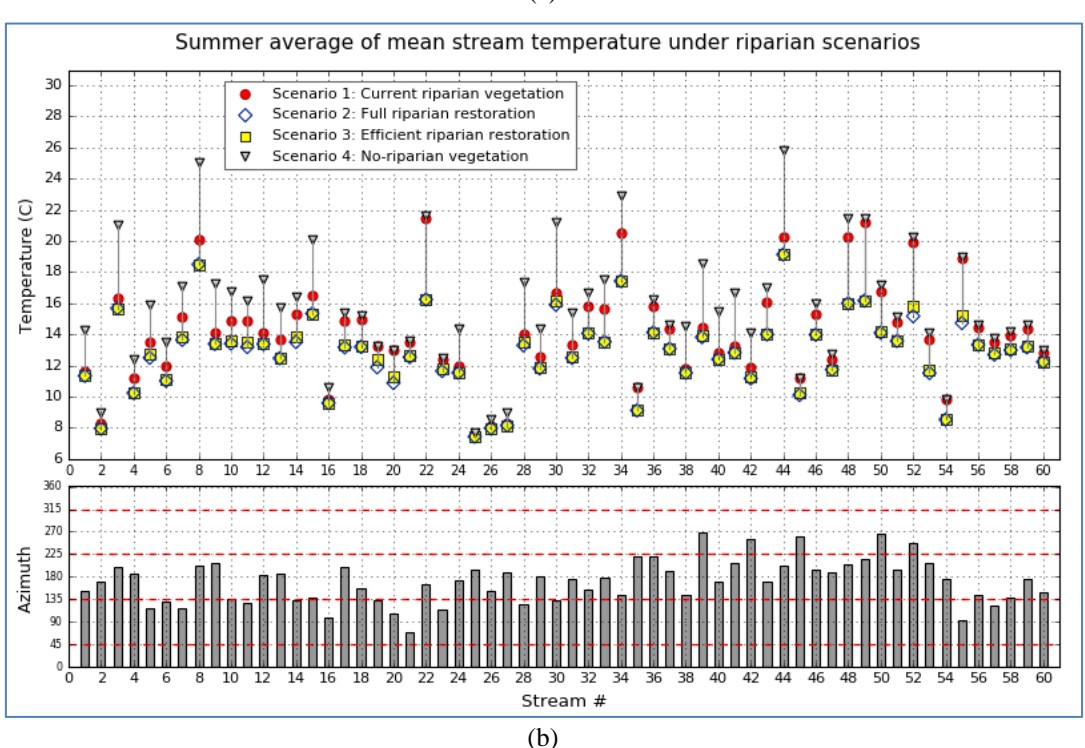

(b)

**Figure 6. Result of mean stream temperature simulation for scenarios 1, 2, and 3. (a) Annual average of stream temperature. (b) Summer average of stream temperature.**

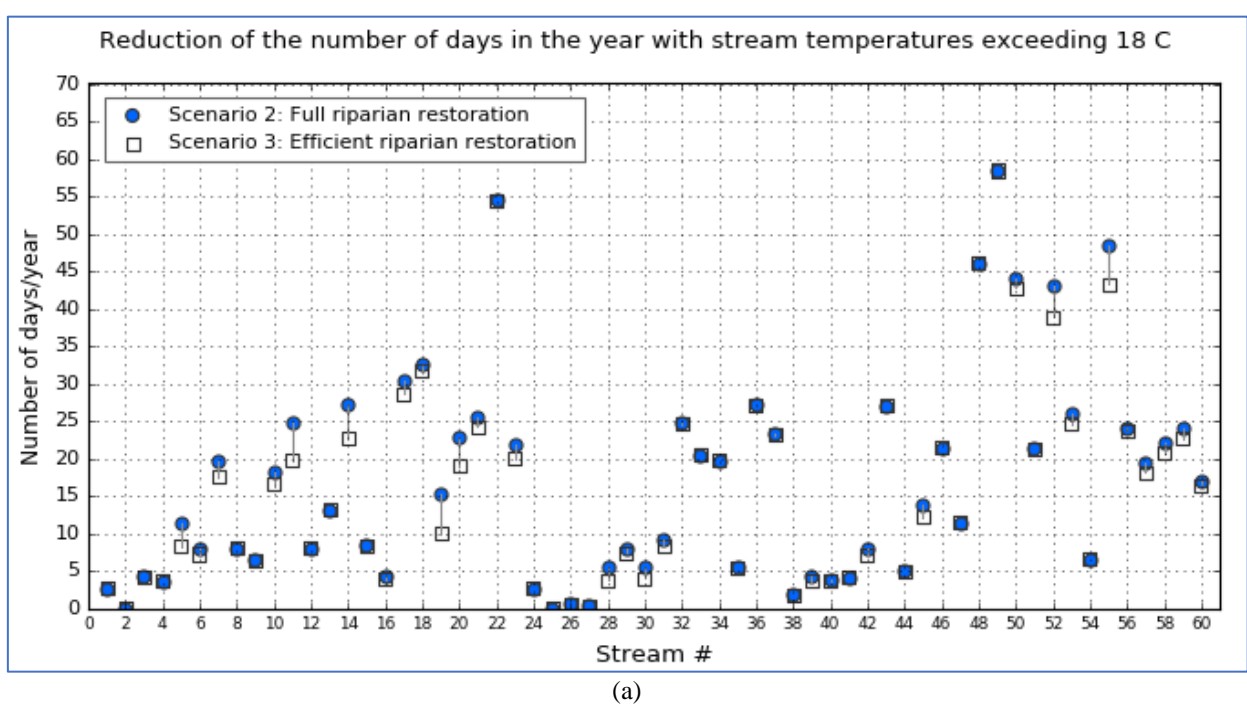

(a)

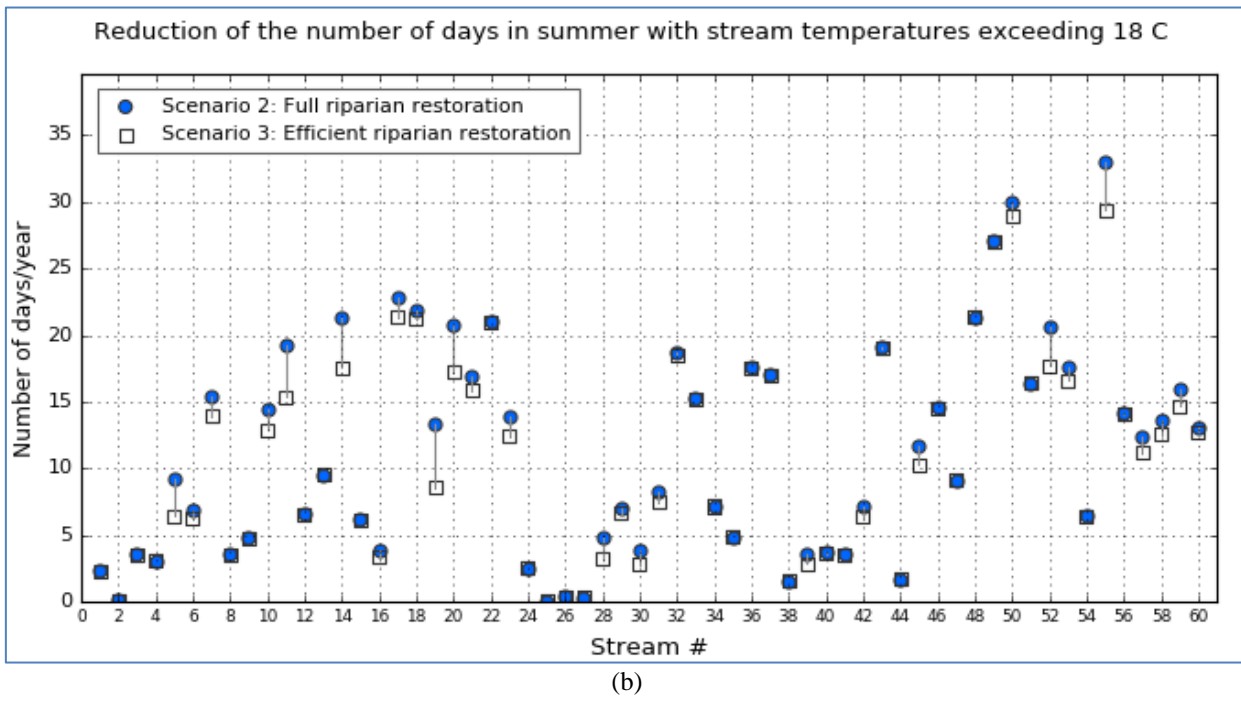

(b)

**Figure 7. Reduction of number of days in (a) the year and in (b) summer with 7dAM stream temperatures exceeding 18 °C in Scenarios 2 and 3.**

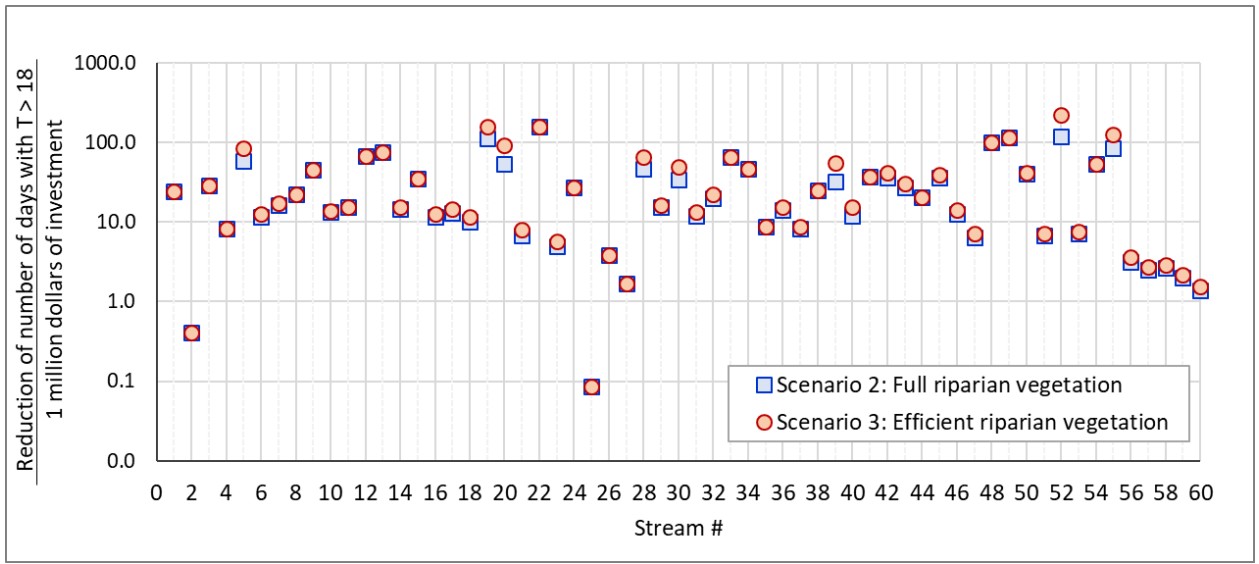

**Figure 8. Benefit-Cost Ratio (Reduction of the number of days that exceed 18 °C / cost of the riparian restoration in millions of dollars) for the 60 DMW stream for the case of full riparian restoration (Scenario 2) and efficient riparian restoration (Scenario 3).**

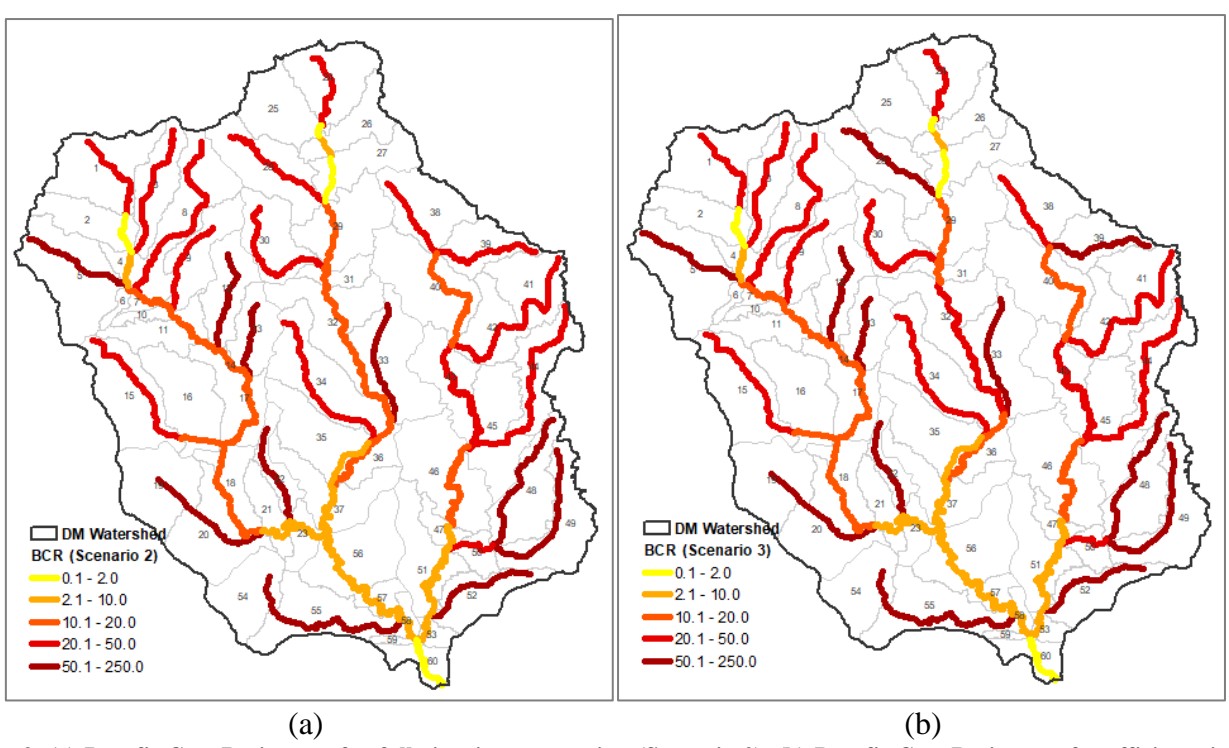

(a)                                      (b)

**Figure 9. (a) Benefit-Cost Ratio map for full riparian restoration (Scenario 2). (b) Benefit-Cost Ratio map for efficient riparian restoration (Scenario 3). The reduction in the number of days that exceed 18 ° C was considered a riparian restoration benefit. The cost corresponds to the 2010 riparian restoration cost in millions of dollars.**

**Table 1. Calibration Coefficients for the Linear, Original Ficklin et al., and Modified Ficklin et al. Stream Temperature Model**

| Calibration site | Modified Ficklin et al. stream temperature model | | | Original Ficklin et al. stream temperature model | | | Linear stream temperature model | | |
|---|---|---|---|---|---|---|---|---|---|
| | NSE | PBIAS | MAE | NSE | PBIAS | MAE | NSE | PBIAS | MAE |
| Sub-basin #31 | 0.74 | -8.2% | 1.65 | 0.77 | -3.6% | 1.41 | 0.46 | 22.8% | 2.47 |
| Sub-basin #59 | 0.82 | -4.4% | 1.40 | 0.85 | -3.1% | 1.31 | 0.7 | 20.4% | 2.28 |

915

920