# Peer review of "An Improved Model of Shade-affected Stream Temperature in Soil & Water Assessment Tool"

_Hydrology and Earth System Sciences, 2022_

## Referee Comment (RC1)

Reviewer: The content is very relevant, as high temperatures in the stream may lead to severe impacts on the river's biota. This new approach seems to be a very efficient manner to estimate the stream temperature, although some uncertainty in this methodology may be still relevant. Additionally, the English of the manuscript seems to be very good, congratulations. Here below and attached are some recommendations.

**Introduction**

In the introduction section you started discussing the main impacts of the increase of stream temperature in the water chemistry (dissolved oxygen, salinity, and pH.), and how these changes may influence certain species, which is totally fair. However, you did not discuss how those changes may affect some hydrological and meteorological parameters, it is essential to put some attention in this discussion as well. See in Koh et al. (2010), Guenther et al. (2012), Dugdale et al. (2018), and Rodrigues et al. (2021). These references discuss some of the main impacts of riparian vegetation in hydrological and meteorological parameters. In addition, you should discuss more other benefits of riparian vegetation, for example, reduce/prevent of river siltation, decrease/alleviate the runoff from the rainfall, reduce of nutrients that go into the stream, etc.

Koh, I., Kim, S., & Lee, D. (2010). Effects of bibosoop plantation on wind speed, humidity, and evaporation in a traditional agricultural landscape of Korea: field measurements and modeling. Agriculture, ecosystems & environment, 135(4), 294-303.

Guenther, S. M., Moore, R. D., & Gomi, T. (2012). Riparian microclimate and evaporation from a coastal headwater stream, and their response to partial-retention forest harvesting. Agricultural and Forest Meteorology, 164, 1-9.

Dugdale, S. J., Malcolm, I. A., Kantola, K., & Hannah, D. M. (2018). Stream temperature under contrasting riparian forest cover: Understanding thermal dynamics and heat exchange processes. Science of the Total Environment, 610, 1375-1389.

Rodrigues, I. S., Costa, C. A. G., Raabe, A., Medeiros, P. H. A., & de Araújo, J. C. (2021). Evaporation in Brazilian dryland reservoirs: Spatial variability and impact of riparian vegetation. Science of The Total Environment, 797, 149059.

I noticed that you separated the objectives from the introduction, which most of the time they are together (Introduction and in the end the objectives). I do not know if it is a requirement of the journal, if so, that's fine you can leave with this separation, if it is not, I would suggest you

incorporate the objectives in the end of the introduction as it is easer for the reader to understand what will be done based in the problem/gap you just mentioned in the introduction.

In addition, I would suggest you reformulate your main objective, as in the end of the introduction you mentioned "…we fill that gap by incorporating the shade factor into the equilibrium temperature approach (Edinger et al., 1974), and couple it with the improved hydroclimatological SWAT model of (Ficklin et al., 2012) to improve the simulation of the heat transfer process at the water-air interface.", if I understood correctly you will include another parameter (shade factor) into the equilibrium temperature approach by Edinger et al. (1974), and then insert it in the hydroclimatological SWAT model. I think you named the "equilibrium temperature approach" to "Energy balance model" in the objective sections, am I right? Please, if it is not required merge the objectives with the introduction.

Why did you choose to improve the Edinger et al. (1974) equilibrium temperature approach? Just asking because there are others energy balance models in the literature…

Why did you just evaluate the reduction in the number of days above survival limits for salmon and trout? Rather than other animals or local flora…

I also do not think you should outline the whole work, like in Lines 109 to 113, it can be removed.

**Methodology**

In Figure 1, I would recommend having a zoom out with the whole map of the USA, before the Oregon map, in fact you could substitute the Oregon map and put both tighter (USA and Oregon). As you are submitting for an international journal, sometimes people do not know where all states from US are located.

In the Line 131 you mentioned that 40% of the area has agriculture, how far these zones are from the study rivers? I was just wondering how you separated the natural riparian vegetation and agriculture areas, which I suppose seems to be the same from above (green). And how the agricultural zones may affect the stream temperature?

Supposing you collected the riparian vegetation data from remote sensing sources (satellite image, for example Landsat time series), how did you differentiate the riparian vegetation from small shrubs as they look almost the same in a pixel of 900 $m^2$?

For the hydrological model, why did you choose SWAT? Please, you have to mention it in your section 2.1

I also noticed that Ficklin et al. (2012) model uses different types of data (snowmelt flow/melt temperature, groundwater temperature, surface runoff, and lateral flow) to compute the local stream temperature. It is fantastic that you have this kind of data in your study area, although I do not think that is so common in remote areas. Do you think it is possible to use the Reanalysis data from ERA 5 data base? I am asking because these types of data are not measured as yours, so it might be a problem. What do you think?

In section 2.4, you mentioned that the shade factor varied from 0 to 1, how and why did you put this range? Is there any reference that mentioned such this nomenclature "shade factor"?

In addition, you also said in Lines 240 and 241 "the shade factor was different for each stream, each day within the year, and each instant within the day." How and in which locations did you calculate the solar radiation for these streams?

**Results**

Section 3.1: It would be great if the NSE, and PBIAS were first explained in the Methodology section, in a subsection called "Statistical Analysis" or something similar.

In section 3.3 you evaluated (very shallow by the way) the effects of riparian vegetation in the stream temperature. Do you think it is necessary to introduce more riparian vegetation in this catchment? I am asking because, although the riparian vegetation can reduce the stream temperature, this may result in other consequences, such as the increase of the transpiration from this vegetation, did you think about that? In addition, I would recommend an additional paragraph for this section 3.3, to discuss more about these consequences.

The section 3.3.2 is very interesting, you should dig more in the literature and discuss more your results with other author, and not only say as you did in Lines 339 and 341 "This finding

is consistent with previous studies seeking strategic placement of riparian vegetation to achieve the greatest reduction in water temperature", please, rephrase this sentence and show the results of these authors and others. This is one of the main important parts of your work, go deeply and find more researches to compare, and discuss. In the end of this part, you should insert something regarding the impacts of the stream temperature in the salmon and trout rearing migration (which until now I do not know why only these two species, I hope I have a suitable justification for this), to make a link with the next section, as you will start to discuss more about soon.

The section 3.3.3, in my opinion, is your peace of gold result, you showed the positive impact of riparian vegetation in reducing the stream temperature, and the reduction in the number of days with exceeding 18 Celsius degree. This is fantastic, and you should discuss these results with other authors, the main benefits (reduction of evaporation, suitable areas for fauna reproduction, etc.) as well as other consequences (this reduction of stream temperature may be excellent for some species, but for others may generate badly effects, please discuss the disadvantages/downside as well). If you can not discuss/compare the number of the days (as I think this result is pretty new), show how this temperature mitigation/attenuation may impact in the local flora, fauna, hydrology, and the meteorology parameters (positive and negative effects).

Regarding section 3.3.4, if possible, do you think would be a good idea to insert the riparian vegetation through the whole river section? Why? Which impacts it would be expected? Please, discuss this situation as well in this section. It could start like "If possible to insert riparian vegetation through the entire river stretch, the main expected impacts would be …"

**Conclusion**

I suggest to change or exclude Lines 391 and 392 when you said "Therefore, the application of the improved stream temperature model can be easily scaled to other regions.", you calibrated this parameter for this specific area, with specific vegetation, as well as meteorological conditions, so be careful.

It is essential a section of "Sources of uncertainty" or "Uncertainty analysis", although the model was well calibrated some uncertainties remain, for example, you calibrated for a period lower than 30 years (for both points of calibration), which may present some bias if in this short

time happened a high or too low discharge; your calibration was specific for this area and for this type of vegetation, how could other researchers use your shade parameter in other areas? Is it really representative? The objective of this section is an auto critique of your own work, showing its limitations and it can be improved. No work is so perfect that can't show any limitations, please rethink about the main uncertainties and write this section.

Overall, the work is very good, it just needs more discussion and some explanations, as I included in the review. The writing is also fine and formal.

---

## Author Comment (AC4)

**Responses to comments and suggestions**

**3. Response to comments and suggestions from the third reviewer:** RC2: **'Comment on hess-2022-116', Anonymous Referee #2, 16 Sep 2022**

Q23

**The introduction is well done, but I think more citations are needed to back up sentences and arguments…**

Citations were added to the introduction, including on the lines mentioned (Lines 47 – 60).

Q24

**I am not sure what the authors mean by "restriction of the watershed hydrological process" on lines 58-59.**

The mentioned sentence (Line 68 in the revised manuscript) was modified by:

"Although statistical models may yield reliable outcomes with few parameters and simple equations (Benyahya et al., 2007; Mohseni & Stefan, 1999), they often do not take into account the right physical structures that characterize the hydrological process and do not take into account the proper interaction of the hydrological process variables (Boyd & Kasper, 2003; Kim & Chapra, 1997)"

Q25

**It may be beneficial for the reader to understand the importance of stream temperature for the DMW. What are the major aquatic species in the DMW that might be influenced by higher stream temperatures? Have there been any fish kills, etc. in the past? I realize that this is a more of a model development paper but including this information will also help understand the application.**

The following sentences were included/added to the article:

In line 37: "For example, in the summer of 2015, the Oregon Department of Fish and Wildlife estimated an approximately 55% reduction in the sockeye salmon population along the lower Columbia River stretch due to stream temperature rising to 24.5 °C (Nguyen, 2021; Sherwood, 2015). Over the past 70 years, the abundance of species such as Coho salmon has shown a drastic decline in California, with similar but less drastic trends in Oregon, due to various factors including elevated stream temperatures (NMFS, 2012, 2014)."

In line 42: "In the local area, Winter Steelhead, Coho Salmon, and resident Cutthroat Trout are among the primary inhabitants of the Dairy McKay watershed streams, whose population is declining due to a variety of water quality factors, including water temperature (CWL, 2019; Hennings, 2014; ODA, 2018). In this regard, the Oregon Plan identified salmon health as a crucial indicator of the ecosystem (Hawksworth, 1999; ODEQ, 2001, 2008, 2010). Additionally, in this area, declines in ecosystem structure and function have also been linked to declines in salmon numbers(Hennings, 2014; ODA, 2018)."

Q26

**I am not sure what "…overlying as far as possible on 12-digit HUC boundaries…"**

HUC stands for Hydrologic Unit Code, which is the US hierarchical watershed classification system. This sentence was clarified in the article by (line 174):

"…overlying as far as possible on 12-digit HUC (Hydrologic Unit Code) boundaries from DMW, which is the US hierarchical watershed classification system."

Q27

**Did the authors examine the influence of tile drainage on stream temperature?**

Tile drainage has not been part of the of the study. However, the approach presented in this study can also be used to assess the effects of tile drainage on stream temperature. Tile drainage directly influences lateral flow and groundwater which are variables within the Ficklin et al. temperature model that was also improved in this work.

Q28

**Include the Forest Grove weather station on Figure 1.**

It was added to Figure 1 as suggested.

Q29

**The calibration and validation procedure for streamflow and stream temperature described the Results and Discussion should be part of the Methods section, as these are not results. Additionally, the streamflow was calibrated using SWAT-CUP, but how was the stream temperature calibrated? Manually?**

Settings for model calibration were moved to the methodology section as "Model Calibration Setup" (section 2.5) (Line 295). However, the calibration results, which also include the evaluation of the shade factor, were kept in the results section (Line 331 and 337).

Regarding the stream temperature calibration, in line 304, the following sentences were included/added in the section 2.5 "Model Calibration Setup"

"The calibration was accomplished by an iterative procedure that was systematized in Python code following the steps shown in table S4 and S5 in section S6 in the Supplement. The Python code to iteratively run SWAT, the input data, required SWAT files, and the modified SWAT model (in Fortran) may be found in the Zenodo repository repository at https://doi.org/10.5281/zenodo.6301709 (Noa-Yarasca, 2022)."

Q30

**Remove the word "In" at the beginning of the sentence on line 178.**

It was removed as recommended (Line 213 in the revised manuscript).

Q31

**…how did the authors implement a 30m buffer around the stream when individual HRU types might be distributed throughout the subbasin? If this is indeed the case, did the different riparian scenarios influence the hydrologic results?**

Riparian vegetation shade and shade factor were calculated independently of the HRUs in a separate GIS environment from the SWAT model. The pre-computed shade factor values were set up as a table that was read by the SWAT model code that was modified for this study. Two new modules were implemented in the SWAT model code to calculate the water temperature following the approach proposed in this work.

The supplement S2 "Shade Factor Calculation", accompanying the article, shows details and equations to compute the shade factor.

All the modified codes, input data, and the code of the modified SWAT model are available in the Zenodo repository at https://doi.org/10.5281/zenodo.6301709 (Noa-Yarasca, 2022).

Q32

**PBIAS should be defined**

In line 311, the section 2.6 "Model Calibration Evaluation" that includes PBIAS definition was added to the article.

Q33

**Section 3.2.1. Why would the shade factor in winter be greater than in the summer? More explanation on this would be beneficial.**

Because of declination angle changes, the solar angle is lower in winter than in summer, resulting in greater projection on the stream and, as a result, more shadow on the stream. This may be the reason why winter shade factor is greater than summer shade factor. When considering vegetation that doesn't keep its leaves throughout the four seasons of the year, this could vary.

Note that this work considered conifers as riparian vegetation, which are common in the DMW Oregon. The majority of these conifers are evergreen, which means they retain most of their leaves all year.

More details on the variation of SF during the year and during the day can be found in the supplement S4 "Shade Factor Temporal Variation" accompanying the article.

Q34

**This is minor, but the C1 and C2 parameters discussed in the Results are presented as c1 and c2 in the methods.**

They were changed as recommended. They are all now $C_1$ and $C_2$. (Line 260, 261, 308, 309, 310, and 360)

Q35

**I would consider using mean absolute error (MAE) in addition to NSE for stream temperature calibration**

MAE was defined in line 325 and MAE values included/added to the Table 1 "Calibration Coefficients for the Linear, Original Ficklin et al., and Modified Ficklin et al. Stream Temperature Model"

Q36

**In Figure 3A it seems that the modified model (and probably the Ficklin model too) has issues with simulating the stream temperature during the winter. I was wondering if the authors could comment on this.**

Although the calibration of the modified stream temperature model achieved encouraging results throughout the year, during the winter the gap between observed and simulated values is notable compared to other periods. As mentioned in the section "**Model limitations and uncertainties**" (Line 492), the model outcomes may be affected by unknowns related to input data, model structure, and model parameters, which may be amplified in the winter. In relation to riparian vegetation, for example, the density of the leaves, which we assumed to be constant, could be playing a role in the model. A lower density of leaves in winter would allow the passage of more solar radiation than a dense canopy that could increase the stream temperature. The energy balance of water during the winter months may also be significantly influenced by additional factors or variables, including hyporheic flow, heat from winter precipitation, heat from bottom friction in winter flows, and others. It is recommended that future research take these aspects into account.

In this regard, in line 365, the following sentences were included in section 3.2.2 "calibration"

"Although the calibration achieved encouraging evaluation coefficients, the gap between observed and simulated values during the winter at the upstream control point (sub-basin #31) is notable compared to other periods. In this period and zone, the stream temperature may be influenced by additional factors or variables that have not been considered in this study. These factors can be, for example, canopy density changes in winter, hyporheic flow, heat from winter precipitation, bottom friction heat in winter flows, and others. Future research is recommended to take these aspects into account."

**References.**

Benyahya, L., Caissie, D., St-Hilaire, A., Ouarda, T. B. M. J., & Bobée, B. (2007). A Review of Statistical Water Temperature Models. *Canadian Water Resources Journal*, *32*(3), 179–192. https://doi.org/10.4296/cwrj3203179

Boyd, M., & Kasper, B. (2003). *Analytical Methods for Dynamic Open Channel Heat and Mass Transfer: Methodology for the Heat Source Model Version 7.0*. http://www.deq.state.or.us/wq/TMDLs/tools.htm

CWL. (2019). *TUALATIN RIVER, TOTAL MAXIMUM DAILY LOAD Implementaion Plan City of West Linn , Oregon City of West Linn* (Issue August 2003).

Hawksworth, J. (1999). *Dairy-McKay Watershed Analysis*. https://www.blm.gov/or/districts/salem/plans/files/watershed_analyses/sdo_dairy_mckay_wa/sdo_dairy_mckay_wachap_1.pdf

Hennings, R. B. (2014). *Stream Temperature Management in the Tualatin Watershed : Is it Improving Salmonid Habitat ?* [Portland State University]. https://pdxscholar.library.pdx.edu/geog_masterpapers/8

Kim, K., & Chapra, S. (1997). Temperature model for highly transient shallow streams. *Journal of Hydraulic Engineering*, *123*(1), 30–40. https://ascelibrary.org/doi/abs/10.1061/(ASCE)0733-9429(1997)123:1(30)

Mohseni, O., & Stefan, H. G. (1999). Stream temperature/air temperature relationship: A physical interpretation. *Journal of Hydrology*, *218*(3–4), 128–141. https://doi.org/10.1016/S0022-1694(99)00034-7

Nguyen, D. (2021). *Warming rivers in US West killing fish, imperiling industry*. AP NEWS. https://apnews.com/article/business-environment-and-nature-fish-climate-change-5c85e86a2ba18171ca55d5de8f89dea3

NMFS. (2012). *Recovery plan for the evolutionarily significant unit of Central California Coast Coho salmon. Volume 1*.

NMFS. (2014). *Final recovery plan for the Southern Oregon/Northern California Coast evolutionarily significant unit of Coho salmon (Oncorhynchus kisutch)*.

Noa-Yarasca, E. (2022). *Data on An Improved Model of Shade-affected Stream Temperature in Soil & Water Assessment Tool* (https://doi.org/10.5281/zenodo.6301709). zenodo.org.

ODA. (2018). *Tualatin River Watershed Agricultural Water Quality Management Area Plan*. www.oregon.gov/ODA/programs/NaturalResources/Pages/Default.aspx

ODEQ. (2001). *Tualatin subbasin Total Maximum Daily Load (TMDL)* (Issue August).

ODEQ. (2008). *Temperature Water Quality Standard Implementation – A DEQ Internal Management Directive*.

ODEQ. (2010). *Cost Estimate to Restore Riparian Forest Buffers and Improve Stream Habitat in the Willamette Basin, Oregon*.
https://www.co.benton.or.us/sites/default/files/fileattachments/community_development/page/2516/willametteripcost030310.pdf

Sherwood, C. (2015). *Thousands of salmon die in hotter-than-usual Northwest rivers*. Reuters.
https://www.reuters.com/article/us-usa-oregon-salmon/thousands-of-salmon-die-in-hotter-than-usual-northwest-rivers-idUSKCN0Q203P20150728

---

## Author Comment (AC5)

**Responses to comments and suggestions**

4. **Response to comments and suggestions from the fourth reviewer:** EC1**: 'Comment on hess-2022-116', Carla Ferreira, 21 Oct 2022**

Q37

**Section 1.1.: please, include this section in the main section.**

Comments and suggestions were considered, included, and added to the introduction and other corresponding sections in the revised article.

Q38

**Section 2.1: please include more information about the hydrological network, the agriculture (e.g. irrigation and main crops), water uses, describe the current condition of the riparian vegetation, etc., so that the reader has a better overview of what is being considered in the model.**

- All the suggestions were included in the revised article.
- All the modified codes, input data, and the code of the modified SWAT model are available in the Zenodo repository at https://doi.org/10.5281/zenodo.6301709 (Noa-Yarasca, 2022).

- Tables of cost of restoration were added in the supplement S7 "Cost of riparian reforestation/restoration for both scenarios: Full riparian and efficient restoration"

Q41

**L129: "temperatures remain degraded" – what do you mean?**

This means that stream temperature in a significant number of DMW streams remain above natural values. The mentioned statement was re-worded and better explained as follows (Line 155):

"Despite improvements in DO levels in certain streams, temperatures in a significant number of streams remain above natural values (CWL, 2019; ODA, 2018)."

Q42

**L125-128: I suggest to present this information after describing the land use (end of this section)**

It was modified as suggested (Line 153 in the revised manuscript)

**L134: Please, correct numbering of the sub-section**

It was modified as suggested (Line 165 in the revised manuscript)

**L143: please, add information about the resolution of the DEM**

It was added as suggested (Line 178 in the revised manuscript)

**L178: delete "In"**

It was deleted as suggested (Line 213 in the revised manuscript)

**Fig. 1: scale bar is missing**

Scale bars were added in Figure 1

[Figure]

**References.**

CWL. (2019). *TUALATIN RIVER, TOTAL MAXIMUM DAILY LOAD Implementation Plan City of West Linn , Oregon City of West Linn* (Issue August 2003).

Noa-Yarasca, E. (2022). *Data on An Improved Model of Shade-affected Stream Temperature in Soil & Water Assessment Tool* (https://doi.org/10.5281/zenodo.6301709). zenodo.org.

ODA. (2018). *Tualatin River Watershed Agricultural Water Quality Management Area Plan*. www.oregon.gov/ODA/programs/NaturalResources/Pages/Default.aspx

---

## Author Response (AR1)

**Author responses to reviewers**

Responses to reviewers' comments and suggestions were organized as follows: (1) Rewritten comments from submitters and the public, (2) Response from authors, and (3) Changes implemented in the revised version of the manuscript. Each reviewer's comment was numbered with the letter "C" followed by a sequential number only to maintain a sequence of the authors' comments.

1. **Response to comments and suggestions from the first reviewer:** RC1**: 'Comment on hess-2022-116', Anonymous Referee #1, 16 Sep 2022**

**C1. "In the introduction section you started discussing the main impacts of the increase of stream temperature in the water chemistry (dissolved oxygen, salinity, and pH.), and how these changes may influence certain species, which is totally fair. However, you did not discuss how those changes may affect some hydrological and meteorological parameters…"**

Statements and paragraphs were added to the introduction and discussion section expanding on the effects of stream temperature on hydrologic parameters as well as additional effects and benefits of riparian vegetation. Here are some of these sentences.

- **"How changes in stream temperature affect hydrological and meteorological parameters?"**

The following was added/included to line 33 of the revised manuscript:

Changes in water temperature also influence hydrological parameters such as evaporation through altering the heat flux at the air-water interface, as well as other parameters indirectly, because all processes in the water cycle are linked (Edinger et al., 1974).

- **Impacts of riparian vegetation in hydrological and meteorological parameters.**
- **The benefits of riparian vegetation are also sediment reduction, alleviating runoff, and nutrients.**

In the line 112 (introduction section) it was added/included:

"Riparian vegetation has been identified as an efficient strategy to control stream temperatures by blocking solar radiation from reaching streams (Chen, Carsel, et al., 1998; Roth et al., 2010; Rutherford et al., 1997). Previous studies in, for example, the US (Abbott G., 2002; Abdi et al., 2020; Chen, McCutcheon, et al., 1998), Brazil (Ishikawa et al., 2021), Europe (Johnson & Wilby, 2015; Kalny et al., 2017; Kałuza et al., 2020), Asia (Liao et al., 2014; Liu et al., 2019), New Zeeland (Rutherford et al., 1997), among other places have demonstrated the efficacy of riparian vegetation restoration in lowering stream temperatures. Riparian vegetation has also been shown to be effective in lowering silt, nutrients, and boosting biodiversity (Malkinson & Wittenberg, 2007; Poole & Berman, 2000). Furthermore, riparian vegetation also impacts hydrological and meteorological parameters. Prior research, for example, found that riparian plants like bibosoop helped to reduce wind speed and evapotranspiration in crop fields in Korean locations (Koh et al., 2010). Guenther et al. (2012) reported effects of logging on vapor pressure, wind speed, and evaporation. Rodrigues et al. (2021) also provided facts about the impact of riparian vegetation on the evaporation of reservoirs. Dugdale et al. (2018) linked riparian vegetation to changes in the flow of energy across the air-water interface and then to evaporation."

In line 462 (Results and Discussion), the section 3.3.5 "**Evaluating additional effects of riparian vegetation for optimal restoration (future research)**" was also aggregated.

**C2. "I noticed that you separated the objectives from the introduction, which most of the time they are together (Introduction and in the end the objectives). I do not know if it is a requirement of the journal, ..."**

The objective was joined to the introduction section and revised and reformulated in accordance with the suggestions.

**C3. "Why did you choose to improve the Edinger et al. (1974) equilibrium temperature approach? Just asking…"**

The energy balance equation involving sources such as long- and short-wave radiation includes implicit quartic terms that make it difficult to manipulate. The quartic terms in the equilibrium equation approach are linearized fairly accurately in the range -30 to 50 C, which is a wide range for typical river water temperatures.

**C4. "Why did you just evaluate the reduction in the number of days above survival limits for salmon and trout? Rather than other animals…"**

Regarding this question, in line 42 the following was added/included:

"In the local area, Winter Steelhead, Coho Salmon, and resident Cutthroat Trout are among the primary inhabitants of the Dairy McKay watershed streams, whose population is declining due to a variety of water quality factors, including water temperature (CWL, 2019; Hennings, 2014; ODA, 2018). In this regard, the Oregon Plan identified salmon health as a crucial indicator of the ecosystem (Hawksworth, 1999; ODEQ, 2001, 2008, 2010). Additionally, in this area, declines in ecosystem structure and function have also been linked to declines in salmon numbers(Hennings, 2014; ODA, 2018)."

**C5. "I also do not think you should outline the whole work, like in Lines 109 to 113, it can be removed."**

Changes were made as recommended. Lines 109 to 113 were removed in the revised version.

**C6. "In Figure 1, I would recommend having a zoom out with the whole map of the USA…"**

Changes were made as recommended. The US map was added to Figure 1 as follows.

[Figure]

Figure 1: Left, Streams, sub-watersheds, and political boundaries of the Dairy McKay Watershed (DMW) (HUC10-1719001003). Top right, location of DMW in the Tualatin River basin, and Bottom right, location of the Tualatin River basin in North-western Oregon, USA.

**C7. "In the Line 131 you mentioned that 40% of the area has agriculture, how far these zones are from the study rivers? I was just wondering how you separated the natural riparian vegetation…"**

In the downstream DMW, streams flow through agricultural areas. Agricultural fields are separated from streams by a buffer of 25 to 30 meters. Previous research has found that riparian buffers with widths of 30 to 50 m provide great advantages in managing stream temperatures (DeWalle, 2010). Thus, we considered taking 30 m of buffer in this study. Existing Vegetation Height (EVH) data retrieved from the Land-fire Program (LP) database (LANDFIRE, 2019) over this buffer zone were averaged and then used to calculate the shade factor (parameters that represented riparian vegetation in the stream temperature model). To simplify, the average height has been considered as a solid barrier that blocks solar radiation.

Agricultural lands separated from rivers by a buffer and composed of short-growing plants would not have a role in limiting solar radiation heading toward streams. However, agricultural lands have a key role in other components of the hydrological cycle such as evapotranspiration, surface flow, and groundwater, all of which are incorporated in the SWAT model and also in the Ficklin stream temperature sub-model. In this way, the stream temperature model takes agricultural areas into consideration indirectly.

Without identifying the type of vegetation, the existing vegetation height data obtained in raster format was averaged and used to calculate the shade factor. This simplification was adopted due to the limited data we faced in the DMW. Further study might add factors such as plant species and canopy density in the shadow factor calculation and examine their influence on stream temperature using the enhanced stream temperature model proposed here.

**C8. For the hydrological model, why did you choose SWAT? Please, you have to mention it in your section 2.1**

The SWAT model, which is based on physical principles, is widely used to evaluate the effects of land-use changes, strategic conservation practices, and non-point sources on flow and water quality at the sub-basin and river basin levels. With outstanding results in terms of regulating flow, erosion, nitrate and other nutrients, the SWAT model has been applied in several watersheds across the world. However, in simulating stream temperature, the SWAT model still uses the linear equation of Stefan & Preud'homme (Stefan & Preud'homme, 1993), which is very limited in evaluating the effects of land changes on stream temperature.

Sentences supporting the use of SWAT were added to the article in section 2.2. in line 167 the following was added/included:

"The SWAT has been utilized in watershed modelling at the sub-basin level in many places across the world with outstanding results in terms of controlling flow, erosion, nitrate, and other nutrients (Abbaspour et al., 2015; Moriasi et al., 2007). The physical-based SWAT model is widely used to assess the impact of non-point sources, strategic conservation practices, conditions of soil management practices, and changes in land use in large and complex watersheds and predict their effects on flow, production of sediments and chemicals, and instream temperature (Neitsch et al., 2009)."

**C9. "I also noticed that Ficklin et al. (2012) model uses different types of data (snowmelt flow/melt temperature, groundwater temperature, surface runoff, and lateral flow) to compute the local stream temperature…"**

Many variables in the Ficklin et al. model, such as lateral flow, groundwater, and surface runoff, are outcomes of the SWAT model. Since the SWAT model has been successfully used in flow and water quality modeling in several European watersheds (Abbaspour et al., 2015), the improved model developed in this article is a promising tool to extend water quality modeling including stream temperature simulation at the sub-basin level. With regard to data such as groundwater and snow temperature, these are variables that show little variation throughout the year and can be obtained from global models/maps. I'm not very familiar with the ERA5 database. However, watershed modeling with SWAT and the ERA5 database have produced effective results in recent studies (Marcinkowski et al., 2022; Senent-aparicio et al., 2021).

**C10. "In section 2.4, you mentioned that the shade factor varied from 0 to 1, how and why did you put this range? Is there any reference that mentioned such this nomenclature "shade factor"?"**

The shade factor (SF) was computed as the rate of solar radiation blocked by the topography and riparian vegetation (represented by the shaded area in the stream generated by the topography and vegetation of the stream banks) divided by the potential solar radiation that would reach the stream surface without any barrier (represented by the stream surface area) (Boyd & Kasper, 2003). The blocked solar radiation was computed as the shaded area on the stream surface and potential solar radiation was computed as the stream surface area. Thus, the maximum value of SF would be one, when the shaded area is equal to the stream surface (full shaded stream), and the minimum value of SF would be zero, when there is no shadow on the stream surface. This nomenclature was employed, for example, buy Rutherford et.al (Rutherford et al., 1997). Similar ratios have been mentioned by authors such as (Boyd & Kasper, 2003) and Loicq et.al. (Loicq et al., 2018).

Prior studies and references that used shading in stream temperature modeling were included in the paragraph added to the introduction (Line 124). In addition to the section 3.2.1 "Shade Factor", the supplement S2 "Shade Factor Calculation" accompanying the article shows details and equations employed to compute the shade factor. The code written in python is also available at (https://github.com/noayarae/SF_model.git).

**C11. "In addition, you also said in Lines 240 and 241 "the shade factor was different for each stream, each day within the year, and each instant within the day." How and in which locations did you calculate the solar radiation for these streams?"**

The shade factor calculation process was developed in the Python environment (available at: https://github.com/noayarae/SF_model.git) (it is mentioned the article – Line 277) and then input into the SWAT model. More details of this process are available in section S2 in the Supplement material accompanying the article. The modified SWAT model and input data is also available as: Data on An Improved Model of Shade-affected Stream Temperature in Soil & Water Assessment Tool. https://doi.org/10.5281/zenodo.6301709 (Noa-Yarasca, 2022) (it is mentioned the article – Line 568). These data include land cover, soil type, water rights, weather (precipitation, temperature, solar radiation, humidity, and wind speed), flow and stream temperature, and the calibrated DMW SWAT model.

**C12. "Section 3.1: It would be great if the NSE, and PBIAS were first explained in the Methodology section, in a subsection called "Statistical Analysis" or something similar."**

In line 295, Section 2.5 "**Model calibration evaluation**" was added to the article as follows. This section defines the criteria with which the model was evaluated, such as NSE, PBIAS and MAE.

**2.6 Model Calibration Evaluation**

The model's efficiency was assessed using the Nash Sutcliffe efficiency criteria (NSE), which is given by the equation below.

$$NSE = 1 - \frac{\sum_{i=1}^{n}(O_i - S_i)^2}{\sum_{i=1}^{n}(O_i - O_{avg})^2} \qquad (12)$$

Where $O_i$ is the observed value at time $i$, $S_i$ is the modeled value at time $i$, $O_{avg}$ is the mean of observed values. NSE values range from $-\infty$ to 1, with 1 indicating a perfect model with zero prediction error, NSE = 0 indicating a model with predictive power equal to the mean of observed values, and negative values indicating a very severe model error with prediction worse than the mean of observed data. Previous research has classified models with NSE values less than 0.5 as unsatisfactory, models with values more than 0.65 as good, and models with values greater than 0.75 as very good (Moriasi et al., 2007).

In addition, the average tendency of the simulated values to be greater or lower than their observed values were measured by percent bias (PBIAS), given by

$$PBIAS = 100 \frac{\sum_{i=1}^{n}(O_i - S_i)}{\sum_{i=1}^{n}O_i} \qquad (13)$$

Where $O_i$ is the observed value at time $i$, $S_i$ is the modeled value at time $i$. PBIAS has an optimum value of 0, with values close to zero suggesting accurate model simulation. Positive values imply overestimation bias, whereas negative values suggest underestimating bias in the model.

Moreover, model error was performed using the mean absolute error (MAE), given by

$$MAE = \frac{1}{n}\sum_{i=1}^{n}|S_i - O_i| \tag{14}$$

This is an arithmetic average of the absolute errors between paired observed and simulated values. The MAE ranges from 0 to $\infty$. Given that it is a negatively oriented score, models with low MAE are preferable, with MAE = 0 being the ideal model.

**C13. "I would recommend an additional paragraph for this section 3.3, to discuss more about these consequences."**

In line 462, the following paragraph was added to include further explanation of the extra benefits of riparian vegetation.

**3.3.5 Evaluating additional effects of riparian vegetation for optimal restoration (future research)**

In addition to the positive impacts of riparian vegetation on stream temperature reduction revealed here and earlier research (Abbott G., 2002; DeWalle, 2010; Garner et al., 2017; Kalny et al., 2017; Roth et al., 2010; Sahatjian, 2013), other impacts should not be overlooked when evaluating the implementation of buffer vegetation. Riparian vegetation has also been linked to other services such as reducing nutrients in streams caused by agricultural and livestock activity (Groh et al., 2020; Lutz et al., 2020), controlling soil erosion and bank stability (Dickey et al., 2021), and controlling storm runoff by slowing down water contribution to streams, absorbing rainwater, and allowing groundwater recharge, among others (Hawes & Smith, 2005). While water temperature regulation is based on the canopy's capacity to block solar radiation, other riparian-vegetation services are linked to plant functional features such as root absorption capability, root density, and root depth. The efficient restoration of riparian vegetation reported in this work does not necessarily imply effective restoration for other purposes (nutrient reduction, flow, and erosion control), since these other services are related not only to the canopy but also to other plant functional properties(Hawes & Smith, 2005; Malkinson & Wittenberg, 2007).

A riparian buffer consisting of a mix of trees, shrubs, and grasses is much more efficient in removing a broad range of contaminants than a riparian buffer consisting primarily of trees. This is because grasses' shallow and dense roots are excellent in slowing overland flow and trapping sediments, whereas tree roots are good at absorbing nutrients from groundwater, stabilizing banks, and regulating streamflow (Hawes & Smith, 2005). Furthermore, trees provide shade to cool the water, habitat for birds and other wild critters, and falling leaves and branches provide a source of food for wildlife and aquatic animals. Thus, grasses and shrubs can provide services that forests cannot (Parkyn, 2004).

On the other side, fully riparian vegetation restoration may greatly increase transpiration on hot days, resulting in greater water extraction from rivers by plants, which may be temporarily detrimental to sensitive aquatic species (Garner et al., 2017; Hernandez-Santana et al., 2011). Furthermore, heavy shade could affect the population of primary food producers such as periphyton and grazing snails, which are

important oxygen providers for secondary consumers, water quality regulators, home to tiny creatures, and soil moisture reservoirs (Hill et al., 1995; National Park Services, 2020; Schiller et al., 2007).

If riparian vegetation could be planted along the entire length of the river, the main expected impact would be a reduction in nutrients, sediments, overflows, and stream temperature in various measures, as well as changes in certain sub - processes of the water cycle in the river environment such as transpiration and aquifer recharge, among others. Other expected consequences include the loss of some primary food producers, which may affect the food chain near the river. The findings of effective riparian vegetation restoration in this work are centered on a single goal: stream temperature. These results may vary in a multi-objective assessment of riparian vegetation restoration. Further work is encouraged to assess and evaluate the implementation of multi-target riparian vegetation.

**C14. "The section 3.3.2 is very interesting, you should dig more in the literature and discuss more your results with other author,"**

Results from other authors were added to section 3.3.2. In line 413, the following was added to the paragraph.

"This finding is consistent with previous studies seeking strategic placement of riparian vegetation to achieve the greatest reduction in water temperature. DeWalle (2010), for example, discovered that during summer solstice, south bank riparian vegetation in E-W streams produced 70% of total daily shade compared to 30% of north bank on a 40°N stream, while in N-S streams shading from both banks were equivalents. Similarly, Garner et al. (2017), reported that planting on the southernmost bank of Northern Hemisphere streams flowing E-W, NE-SW, or NW-SE, and vice versa, would result in optimal planting targeted at cooling stream water due to its greater contribution in shadowing compared to the northern bank. Likewise, Jackson et al. (2021), found that in E-W/W-E oriented rivers, the contribution of the north bank riparian vegetation was negligible when compared to the south bank. Thus, tree planting on the north side may be unnecessary for stream temperature control. In N-S/E-N oriented streams, the riparian vegetation on both sides had the same shading effect on streams."

**C15. "The section 3.3.3, in my opinion, is your peace of gold result, you showed the positive impact of riparian vegetation in reducing the stream temperature…"**

Results from other authors were added to section 3.3.3. The following paragraph was added in line 434.

"Previous studies have also obtained positive relationships between increased riparian vegetation and reduced stream temperature using various metrics (Abbott G., 2002; Garner et al., 2017; Kalny et al., 2017; Parkyn, 2004; Wondzell et al., 2019). However, given that future climate change scenarios foresee prolonged hot days that would affect aquatic life (Brander, 2007), this work presents the reduction of days with 7dAM that exceed 18°C, which could be a more practical value/metric for experts and non-experts. The reduction in the number of days with 7dAM indicates encouraging findings for DMW; nevertheless, it was not able to compare with earlier research since they directly concentrate on temperature reduction under various conditions."

Positive and negative effects of adding riparian vegetation were also included and discussed in the added section 3.3.5 "Evaluating additional effects of riparian vegetation for optimal restoration (future research)" (Line 462)

**C16. "Do you think would be a good idea to insert the riparian vegetation through the whole river section? Why? Which impacts it would be expected?..."**

The following paragraph was added in line 455 in section 3.3.4 to answer this question.

"If riparian vegetation could be planted along the entire length of the river, the main expected impact would be a reduction in nutrients, sediments, overflows, and stream temperature in various measures, as well as changes in certain sub - processes of the water cycle in the stream environment such as transpiration and aquifer recharge, among others. Other expected consequences include the loss of some primary food producers, which may affect the food chain near the river. The findings of effective riparian vegetation restoration in this work are centered on a single goal: stream temperature. These results may vary in a multi-objective assessment of riparian vegetation restoration. Further work is encouraged to assess and evaluate the implementation of multi-target riparian vegetation."

Additionally, in line 462, the section 3.3.5 "Evaluating additional effects of riparian vegetation for optimal restoration (future research)" was added/included to answer the second part of the question.

**C17. "I suggest to change or exclude Lines 391 and 392 …"**

The indicated sentence was changed as suggested by (Line 554):

"Therefore, the application of the improved stream temperature model could be replicated in other regions with characteristics similar to the DMW"

**C18. "It is essential a section of "Sources of uncertainty" or "Uncertainty analysis"…"**

In line 492, the section 3.3.6 "Model limitations and uncertainties" was added to the article as follows.

**3.3.6 Model limitations and uncertainties**

[revised manuscript text omitted]

**C19. "In the current form of the article, the shade factor and restoration scenarios approach is simplistic and almost decoupled from plant diversity management in the riparian area. I would suggest adding a paragraph in the introduction and perhaps a comparative table about the existing literature …."**

Most of the literature related to the shade factor in the riparian context for stream temperature has been reviewed and added in the different sections of the article, dor example, in line 112 and line 124. Here, I summarized some points to specifically answer this question. These answers are also included/added in the article.

The literature shows a significant number of studies to evaluate the shade of riparian vegetation on streams (Abbott G., 2002; DeWalle, 2010; Fuller et al., 2022; Garner et al., 2017; LeBlanc & Brown, 2000; Li et al., 2012; Loicq et al., 2018; Roth et al., 2010; Wondzell et al., 2019). Models for determining shading or shade factor often included hydraulic and morphological properties of the river, plant characteristics in the buffer zone, and meteorological data such as solar radiation. Complex models, conducted mainly at a local scale (at specific sections of a river or short stretches of a river), have incorporated variables such as canopy shape, canopy overhang, stream bank height, canopy transmittivity, planta species, and others. These complex models also required detailed information at field level on river morphology, detailed canopy features, and in situ meteorological measurements (Davies-Colley et al., 2009; Davies-Colley & Rutherford, 2005; Li et al., 2012). However, in large stretches of rivers where information at the field level is not available yet due to limited resources, simplified models have been employed to determine the shade factor with good enough results (Fuller et al., 2022; Marteau et al., 2022; Seyedhashemi et al., 2022; Spanjer et al., 2022). As mentioned above, accurate assessment of SF has been conducted only at specific points or sections of rivers or short reaches of rivers.

Beyond the calculation of the shading factor, in a broader context of evaluation of the temperature of the stream, no physically based hydrological model has considered the calculation of stream temperature including a detailed mass and energy balance equation that includes riparian vegetation. The challenge of this work is not essentially to improve the accuracy of the SF calculation over existing methodologies, but rather to incorporate the shade component that represents riparian vegetation into a large-scale physically based hydrological model. In this aspect, this study takes a straightforward methodology to determinate the shade factor maintaining the more representative stream and canopy features. When larger and more detailed measurements are available to make a finer calculation of the shade factor in following years, outcomes of the hydrological model might be updated considering this study's approach of incorporating riparian vegetation in the evaluation of stream temperature at the sub-basin and watershed levels.

Notwithstanding, a paragraph indicating the main scope of the SF calculation methodologies was added to the article (line 124).

**C20. "There are only 89 articles for "shade factor" AND riparian on Google Scholar; many are highly relevant to the paper's topic. An analysis of the most relevant ones would provide the reader with an image of the role of plant species and their measurable traits on the shade factor…."**

The main works in the literature that involve the calculation of the shade factor have been reviewed. However, the goal of this work is not to improve the accuracy of the shade factor over existing methodologies, but rather to incorporate the shade factor representing the riparian vegetation into a large-scale physically based hydrological model. The study considered the typical features of Oregonian conifers. Future research should examine how tree species affect shading factor and consequently stream temperature at sub-basin level using the SWAT model.

**C21. "I don't know if your scenarios could be refined in sub-scenarios with different species compositions to test the model's sensitivity to species diversity. This would be extremely valuable for biodiversity management. If it cannot be done now, it could be at least discussed."**

Evaluating different species would involve including variables such as density, transmissivity that are not available for the more common Oregon species (hemlocks, true firs, spruce, Douglas fir and pine, Douglas maple, bigleaf maple, and others). This study considered the general features of long-lived tree species, such as the evergreen forest that is quite common in DMW river buffer zones (ODA, 2018; ODEQ, 2008). As more data is collected, such as tree species and canopy shapes throughout the DMW, more sophisticated shade factor models can be used to assess the effects of plant species on stream temperature at the DMW sub-basin level.

**C22. "The context can then be used in the discussion to analyze the potential cooperation between hydrologists and ecologists for riparian forest management. Riparian vegetation is involved in producing many ecosystem services, not only in water temperature control, and some tradeoffs are between…."**

Added discussion of other riparian vegetation services to article in section 3.3.5 "Evaluating additional effects of riparian vegetation for optimal restoration (future research)" (Line 462)

3. **Response to comments and suggestions from the third reviewer:** RC2: **'Comment on hess-2022-116', Anonymous Referee #2, 16 Sep 2022**

**C23. "The introduction is well done, but I think more citations are needed to back up sentences and arguments…"**

Citations were added to the introduction, including on the lines mentioned (Lines 47 – 60).

**C24. "I am not sure what the authors mean by "restriction of the watershed hydrological process" on lines 58-59."**

The mentioned sentence (Line 68 in the revised manuscript) was modified by:

"Although statistical models may yield reliable outcomes with few parameters and simple equations (Benyahya et al., 2007; Mohseni & Stefan, 1999), they often do not take into account the right physical structures that characterize the hydrological process and do not take into account the proper interaction of the hydrological process variables (Boyd & Kasper, 2003; Kim & Chapra, 1997)"

**C25. "It may be beneficial for the reader to understand the importance of stream temperature for the DMW. What are the major aquatic species in the DMW that might be influenced by higher stream temperatures? Have there been any fish kills, etc. in the past? I realize that this is a more of a model development paper but including this information will also help understand the application."**

The following sentences were included/added to the article:

In line 37: "For example, in the summer of 2015, the Oregon Department of Fish and Wildlife estimated an approximately 55% reduction in the sockeye salmon population along the lower Columbia River stretch due to stream temperature rising to 24.5 °C (Nguyen, 2021; Sherwood, 2015). Over the past 70 years, the abundance of species such as Coho salmon has shown a drastic decline in California, with similar but less drastic trends in Oregon, due to various factors including elevated stream temperatures (NMFS, 2012, 2014)."

In line 42: "In the local area, Winter Steelhead, Coho Salmon, and resident Cutthroat Trout are among the primary inhabitants of the Dairy McKay watershed streams, whose population is declining due to a variety of water quality factors, including water temperature (CWL, 2019; Hennings, 2014; ODA, 2018). In this regard, the Oregon Plan identified salmon health as a crucial indicator of the ecosystem (Hawksworth, 1999; ODEQ, 2001, 2008, 2010). Additionally, in this area, declines in ecosystem structure and function have also been linked to declines in salmon numbers(Hennings, 2014; ODA, 2018)."

**C26. "I am not sure what "…overlying as far as possible on 12-digit HUC boundaries…"…"**

HUC stands for Hydrologic Unit Code, which is the US hierarchical watershed classification system. This sentence was clarified in the article by (line 174):

"…overlying as far as possible on 12-digit HUC (Hydrologic Unit Code) boundaries from DMW, which is the US hierarchical watershed classification system."

**C27. "Did the authors examine the influence of tile drainage on stream temperature?"**

Tile drainage has not been part of the of the study. However, the approach presented in this study can also be used to assess the effects of tile drainage on stream temperature. Tile drainage directly influences lateral flow and groundwater which are variables within the Ficklin et al. temperature model that was also improved in this work.

**C28. "Include the Forest Grove weather station on Figure 1."**

It was added to Figure 1 as suggested.

[Figure]

**Figure 2: Left, Streams, sub-watersheds, and political boundaries of the Dairy McKay Watershed (DMW) (HUC10-1719001003). Top right, location of DMW in the Tualatin River basin, and Bottom right, location of the Tualatin River basin in North-western Oregon, USA.**

**C29. "The calibration and validation procedure for streamflow and stream temperature described the Results and Discussion should be part of the Methods section, as these are not results. Additionally, the streamflow was calibrated using SWAT-CUP, but how was the stream temperature calibrated? Manually?"**

Settings for model calibration were moved to the methodology section as "Model Calibration Setup" (section 2.5) (Line 295). However, the calibration results, which also include the evaluation of the shade factor, were kept in the results section (Line 331 and 337).

Regarding the stream temperature calibration, in line 304, the following sentences were included/added in the section 2.5 "Model Calibration Setup"

"The calibration was accomplished by an iterative procedure that was systematized in Python code following the steps shown in table S4 and S5 in section S6 in the Supplement. The Python code to iteratively run SWAT, the input data, required SWAT files, and the modified SWAT model (in Fortran) may be found in the Zenodo repository repository at https://doi.org/10.5281/zenodo.6301709 (Noa-Yarasca, 2022)."

**C30. "Remove the word "In" at the beginning of the sentence on line 178."**

It was removed as recommended (Line 213 in the revised manuscript).

**C31. "…how did the authors implement a 30m buffer around the stream when individual HRU types might be distributed throughout the subbasin? If this is indeed the case, did the different riparian scenarios influence the hydrologic results?"**

Riparian vegetation shade and shade factor were calculated independently of the HRUs in a separate GIS environment from the SWAT model. The pre-computed shade factor values were set up as a table that was read by the SWAT model code that was modified for this study. Two new modules were implemented in the SWAT model code to calculate the water temperature following the approach proposed in this work.

The supplement S2 "Shade Factor Calculation", accompanying the article, shows details and equations to compute the shade factor.

All the modified codes, input data, and the code of the modified SWAT model are available in the Zenodo repository at https://doi.org/10.5281/zenodo.6301709 (Noa-Yarasca, 2022).

**C32. "PBIAS should be defined"**

In line 311 of the revised manuscript, section 2.6 "Model Calibration Evaluation" that includes PBIAS definition was added to the article.

**C33. "Section 3.2.1. Why would the shade factor in winter be greater than in the summer? More explanation on this would be beneficial."**

Because of declination angle changes, the solar angle is lower in winter than in summer, resulting in greater projection on the stream and, as a result, more shadow on the stream. This may be the reason why winter shade factor is greater than summer shade factor. When considering vegetation that doesn't keep its leaves throughout the four seasons of the year, this could vary.

Note that this work considered conifers as riparian vegetation, which are common in the DMW Oregon. The majority of these conifers are evergreen, which means they retain most of their leaves all year.

More details on the variation of SF during the year and during the day can be found in the supplement S4 "Shade Factor Temporal Variation" accompanying the article.

**C34. "This is minor, but the C1 and C2 parameters discussed in the Results are presented as c1 and c2 in the methods."**

They were changed as recommended. They are all now $C_1$ and $C_2$. (Line 260, 261, 308, 309, 310, and 360)

**C35. "I would consider using mean absolute error (MAE) in addition to NSE for stream temperature calibration"**

MAE was defined in line 325 and MAE values included/added to the Table 1 "Calibration Coefficients for the Linear, Original Ficklin et al., and Modified Ficklin et al. Stream Temperature Model"

**C36. "In Figure 3A it seems that the modified model (and probably the Ficklin model too) has issues with simulating the stream temperature during the winter. I was wondering if the authors could comment on this."**

Although the calibration of the modified stream temperature model achieved encouraging results throughout the year, during the winter the gap between observed and simulated values is notable compared to other periods. As mentioned in the section "**Model limitations and uncertainties**" (Line 492), the model outcomes may be affected by unknowns related to input data, model structure, and model parameters, which may be amplified in the winter. In relation to riparian vegetation, for example, the density of the leaves, which we assumed to be constant, could be playing a role in the model. A lower density of leaves in winter would allow the passage of more solar radiation than a dense canopy that could increase the stream temperature. The energy balance of water during the winter months may also be significantly influenced by additional factors or variables, including hyporheic flow, heat from winter precipitation, heat from bottom friction in winter flows, and others. It is recommended that future research take these aspects into account.

In this regard, in line 365, the following sentences were included in section 3.2.2 "calibration"

"Although the calibration achieved encouraging evaluation coefficients, the gap between observed and simulated values during the winter at the upstream control point (sub-basin #31) is notable compared to other periods. In this period and zone, the stream temperature may be influenced by additional factors or variables that have not been considered in this study. These factors can be, for example, canopy density changes in winter, hyporheic flow, heat from winter precipitation, bottom friction heat in winter flows, and others. Future research is recommended to take these aspects into account."

**4. Response to comments and suggestions from the fourth reviewer:** EC1**: 'Comment on hess-2022-116', Carla Ferreira, 21 Oct 2022**

**C37. "Section 1.1.: please, include this section in the main section."**

Comments and suggestions were considered, included and added to the introduction and other corresponding sections in the article.

**C38. "Section 2.1: please include more information about the hydrological network, the agriculture (e.g. irrigation and main crops), water uses, describe the current condition of the riparian vegetation, etc., so that the reader has a better overview of what is being considered in the model."**

- All the suggestions were included in the paper.
- All the modified codes, input data, and the code of the modified SWAT model are available in the Zenodo repository at https://doi.org/10.5281/zenodo.6301709 (Noa-Yarasca, 2022).

- Tables of cost of restoration were added in the supplement S7 "Cost of riparian reforestation/restoration for both scenarios: Full riparian and efficient restoration"

**C39 & C40. "L25-29: add references" and "L126-128: add references"**

References were added in the corresponding lines.

**C41. "L129: "temperatures remain degraded" – what do you mean?"**

This means that stream temperature in a significant number of DMW streams remain above natural values. The mentioned statement was re-worded and better explained as follows (Line 155):

"Despite improvements in DO levels in certain streams, temperatures in a significant number of streams remain above natural values (CWL, 2019; ODA, 2018)."

**C42. "L125-128: I suggest to present this information after describing the land use (end of this section)"**

It was modified (switch) as follows (Line 153 in the revised manuscript)

145 **2.1 Dairy McKay Watershed Case Study**

The Dairy-McKay watershed (DMW) (Hydrologic Unit Code (HUC)-10: 1709001003), located in Northwestern Oregon, is part of the Tualatin sub-basin (HUC-8: 17090010). It encompasses an area of 598.3 square kilometers draining into the Tualatin River (Fig. 1). The DMW is characterized by higher elevations and varied topography of the Coast Range in the northern part and flat topography in the southern. The highest elevation corresponds to 690 masl, while the lowest one corresponds to 35

150 masl at the confluence with the Tualatin River. Characterized by having perennial flow, DMW is considered one of the main tributaries of the Tualatin River, which is the prominent channel within the watershed. The major area of DMW is located across Washington county (97.4%), and 1.3% across Multnomah, and the last 1.3% across Columbia County.

The DMW climate corresponds to a Mediterranean climate with the lack of rains in summer (51 mm) and mild intensity, long duration rains in winter (719 mm). DMW soils are mainly composed of fine soils such as silt and clay with abundant natural

155 phosphate. Despite improvements in DO levels in certain streams, temperatures in a significant number of streams remain above natural values (CWL, 2019; ODA, 2018). Regarding land use, there are three main areas: the northern half area is dominated by forestry involving around 55% of the DMW, the middle part is dominated by agriculture that encompasses around 40%, and the southern part is dominated by a growing urban area by around 5%. The upstream part of the DMW is dominated by long-lived trees species such as evergreen forest and shrubland, while the downstream part is dominated by

160 seasonal crops such as Slender Wheatgrass, and at the most downstream extent, is dominated by urban areas. Due to the predominance of fine soils, upstream areas are vulnerable to erosion and landslides phenomena (Hawksworth, 1999). In agricultural areas, water quality has been found to degrade rapidly, with higher water temperature and higher phosphorus

concentrations (CWL, 2019; Hawksworth, 1999; ODEQ, 2001). Some streams such as the West Fork Dairy Creek show lower Dissolved Oxygen (DO) levels than natural conditions, limiting aquatic life (Hennings, 2014; ODA, 2018).

**C43. "L134: Please, correct numbering of the sub-section"**

It was modified as suggested (Line 165 in the revised manuscript). The numbering of the section to which it refers was corrected as follows

165 **2.2 Hydrologic Model**

Hydrological processes for DMW were simulated by using the Soil and Water Assessment Tool 2012 (Neitsch et al., 2011); developed by the United States Department of Agriculture (USDA) Agricultural Research Service (ARS). The SWAT has

**C44. "L143: please, add information about the resolution of the DEM"**

It was added as follows (Line 178 in the revised manuscript)

175   (Hydrologic Unit Code) boundaries from DMW, which is the US hierarchical watershed classification system. For modeling purposes, each sub-basin was divided into small areas called "Hydrologic Response Units" (HRU), which are portions of areas that have unique combinations of slope topography, land use, and soil type features. Slope topography was calculated from DEM (cell size 10x10m) and classified in three ranges: 0-5%, 5-20%, and greater than 20%. The land use and soil data were retrieved from the National Land Cover Database (NLCD) and Soil Survey Geographic Database (SSURGO) in raster format

180   with 10x10m cell size (USDA, n.d.). To eliminate small coverage areas of these features into each HRU, a threshold of 10%

**L178: delete "In"**

It was deleted as suggested (Line 213 in the revised manuscript)

**Fig. 1: scale bar is missing**

Scale bars were added in Figure 1

[Figure]

**References for these responses.**

[revised manuscript text omitted]

---

## Author Response (AR2)

**Author responses to the Editor decision: Publish subject to minor revisions (review by editor. By Carla Ferreira)**

Responses to Editor comments and suggestions were organized as follows: (1) Rewritten comments, (2) Response from authors, and (3) Changes implemented in the revised version of the manuscript. Each reviewer's comment was numbered with the letter "C" followed by a sequential number only to maintain a sequence.

**C1. Introduce DO abbreviation in L179 rather than L188.**

Changes were made as recommended (In the revised version, the L179 and L188 correspond to L156 and L166, respectively).

- The DO abbreviation has been set in L156.

| | |
|---|---|
| 155 duration rains in winter (719 mm). DMW soils are mainly composed of fine soils such as silt and clay with abundant natural phosphate. Despite improvements in Dissolved Oxygen (DO) levels in certain streams, temperatures in a significant number of streams remain above natural values (CWL, 2019; ODA, 2018). Regarding land use, there are three main areas: the northern half area is dominated by forestry involving around 55% of the DMW, the middle part is dominated by agriculture that encompasses around 40%, and the southern part is dominated by a growing urban area by around 5%. The upstream part of 160 the DMW is dominated by long-lived trees species such as evergreen forest and shrubland, while the downstream part is dominated by seasonal crops such as Slender Wheatgrass, and at the most downstream extent, is dominated by urban areas. Due to the predominance of fine soils, upstream areas are vulnerable to erosion and landslides phenomena (Hawksworth, | Efrain Noa-Yarasca **Deleted:** DO |
| 1999). In agricultural areas, water quality has been found to degrade rapidly, with higher water temperature and higher 165 phosphorus concentrations (CWL, 2019; Hawksworth, 1999; ODEQ, 2001). Some streams such as the West Fork Dairy Creek show lower DO levels than natural conditions, limiting aquatic life (Hennings, 2014; ODA, 2018).

**2.2 Hydrologic Model** | Efrain Noa-Yarasca **Deleted:** Dissolved Oxygen (

Efrain Noa-Yarasca **Deleted:** ) |

**C2. L396: add reference to Fig. 1**

Changes were made as recommended (In the revised version the L396 corresponds to L365).

- Reference to Fig. 1 has been added.

> **3.2.2 Calibration**
>
> The values of the four calibrated coefficients ($\lambda$, tair lag, $C_1$, and $C_2$) driving the modified stream temperature model were
> 365 0.88, 5, 0.67, and 1.16 for sub-basin #31 and 1.06, 6, 0.74, and 1.17 for sub-basin #59, respectively (Fig. 1). The Nash Sutcliffe Efficiency (NSE) values for sub-basins # 31 and # 59 were 0.74 and 0.82, respectively. These two NSE values are considered as good fit and very good fit (Moriasi et al., 2007), respectively, and are consistent with successful calibrations reported in other studies ranging from 0.70 to 0.89 (Du et al., 2018; Ficklin et al., 2012; Mustafa et al., 2018). Figure 3a-b shows the

**C3. Please, add units for the MAE values throughout the paper and tables.**

Changes were made as recommended.

- In L332, it was clarified that MAE is given in the same units as the target variable (temperature).
- In Table 1, MAE units were added.

Moreover, model error was performed using the mean absolute error (MAE), given by

330    $$MAE = \frac{1}{n}\sum_{i=1}^{n}|S_i - O_i|$$    (14)

This is an arithmetic average of the absolute errors between paired observed and simulated values. The MAE ranges from 0 to ∞ in the same units as the target variable (temperature - °C). Given that it is a negatively oriented score, models with low MAE are preferable, with MAE = 0 being the ideal model.

**3 Results and Discussion**

Table 1. Calibration Coefficients for the Linear, Original Ficklin et al., and Modified Ficklin et al. Stream Temperature Model

| Calibration site | Modified Ficklin et al. stream temperature model | | | Original Ficklin et al. stream temperature model | | | Linear stream temperature model | | |
|---|---|---|---|---|---|---|---|---|---|
| | NSE | PBIAS | MAE (°C) | NSE | PBIAS | MAE (°C) | NSE | PBIAS | MAE (°C) |
| Sub-basin #31 | 0.74 | -8.2% | 1.65 | 0.77 | -3.6% | 1.41 | 0.46 | 22.8% | 2.47 |
| Sub-basin #59 | 0.82 | -4.4% | 1.40 | 0.85 | -3.1% | 1.31 | 0.7 | 20.4% | 2.28 |

Efrain Noa-Yarasca
Formatted Table

920